# A Safety-Aware Location Privacy-Preserving IoV Scheme with Road Congestion-Estimation in Mobile Edge Computing

**DOI:** 10.3390/s23010531

**Published:** 2023-01-03

**Authors:** Messaoud Babaghayou, Noureddine Chaib, Nasreddine Lagraa, Mohamed Amine Ferrag, Leandros Maglaras

**Affiliations:** 1LIM Lab., Amar Telidji University, Laghouat 03000, Algeria; 2Technology Innovation Institute, Masdar City 9639, Abu Dhabi, United Arab Emirates; 3Blockpass ID Lab., Edinburgh Napier University, Edinburgh EH10 5DT, UK

**Keywords:** location tracking, safety and privacy trade-off, road congestion reporting, pseudonym change, location privacy statistics

## Abstract

By leveraging the conventional Vehicular Ad-hoc Networks (VANETs), the Internet of Vehicles (IoV) paradigm has attracted the attention of different research and development bodies. However, IoV deployment is still at stake as many security and privacy issues are looming; location tracking using overheard safety messages is a good example of such issues. In the context of location privacy, many schemes have been deployed to mitigate the adversary’s exploiting abilities. The most appealing schemes are those using the silent period feature, since they provide an acceptable level of privacy. Unfortunately, the cost of silent periods in most schemes is the trade-off between privacy and safety, as these schemes do not consider the timing of silent periods from the perspective of safety. In this paper, and by exploiting the nature of public transport and role vehicles (overseers), we propose a novel location privacy scheme, called OVR, that uses the silent period feature by letting the overseers ensure safety and allowing other vehicles to enter into silence mode, thus enhancing their location privacy. This scheme is inspired by the well-known war strategy “Give up a Pawn to Save a Chariot”. Additionally, the scheme does support road congestion estimation in real time by enabling the estimation locally on their On-Board Units that act as mobile edge servers and deliver these data to a static edge server that is implemented at the cell tower or road-side unit level, which boosts the connectivity and reduces network latencies. When OVR is compared with other schemes in urban and highway models, the overall results show its beneficial use.

## 1. Introduction

The benefits of technology as it has emerged in recent years are innumerable and, in general, have benefited the human sector to a great extent, as technology can provide facilities and amenities that help people’s daily lives at all levels. Unfortunately, there is a downside to all of this technology, which needs even more protection towards the human sector. The key is to use technology at the same time to fully protect privacy and make sure that everyone works in a protected information environment.

Additionally, there is the fact that there are processes that are becoming more and more automated, which require even more attention to the protection of the human factor. For this reason, technology contains many positive elements for the human factor, but it also contains some negative elements that need special attention. The contribution of technology in general is huge in many areas. Nevertheless, a very basic issue is the fact that there is a limit to what technology will achieve because it is constantly evolving and affects every aspect of human life that needs special attention.

### 1.1. Incorporation

The rapid technological development of recent years and the facilities provided by technology products have made it necessary for their daily use by more and more people. Electronic devices such as smartphones, smartwatches, tablets, and smart cars have entered for good into the daily life of the person who, through them, has access to a variety of services. Their user can access their bank account from almost anywhere, check the status of their flight, book a restaurant, chat with friends or even find out if any of them are near them and be informed about the weather conditions that will prevail in the area in the coming hours using various applications and services.

All of the above can happen thanks to a huge range of sensors in these devices, which give them access to some information about their environment. They make it possible to locate the user in real time, thus giving access to services that use “geo-location”, which are called located-based services or LBSs [1]. Numerous applications in the fields of “Direction and Navigation”, “Weather”, “Venue” and even “Social games” use geo-location to offer their services.

### 1.2. Limitation

In order to be able to enjoy the above services, as is obvious, the user must reveal their geographical location at the given time. However, this causes a loss of control over their personal privacy [2] as LBSs can use and store the user’s position and movements in it. The sequence of a user’s movements in a site as well as the time they “spent” in it is called mobility trace. The information contained in the mobility trace of each person is huge and their analysis can give knowledge about the place where they work, their family, their health, their habits, the people they meet and even about their religion or their sexual preferences. This sensitive personal information and its acquisition are a goal for many companies worldwide, as their collection and analysis can bring them profits, either through marketing after becoming aware of the individual’s consumer habits, or by selling them to third parties. In addition, many applications that we use every day collect and store geo-location information with or without our consent, thus violating our personal privacy.

### 1.3. Specification

Smart cars will come with a number of communication protocols (e.g., IEEE 802.11P and IEEE 1609) and capabilities like Vehicle-to-Vehicle (V2V) and Vehicle to Infrastructure (V2I) creating the basis for the promising Internet of Vehicles (IoV) network. IoV can also benefit from the various sensors and embedded systems, which will result in large-scale indirect communications, as illustrated in Figure 1.

A vehicle, using direct communications with other entities, can facilitate road safety enhancement by broadcasting Basic Safety Messages (BSMs), which are called beacons. These BSMs contain fine-grained location data, thus opening privacy-related issues, especially those related to location privacy. A BSM has different fields and packet formats, just like in Figure 2.

### 1.4. Motivation

The emerging privacy-related issues were tackled by the research community in different ways; one of the most common and effective strategies is the “silent period”, which consists of ceasing broadcasting safety messages. Two major drawbacks were produced: blind-sacrificing of safety for the sake of privacy since most of the literature schemes do use the silence period technique without much considerations, secondly, vehicles are invisible from road congestion applications that are enabled for the one-hope V2V communications and hence, invisible vehicles do influence the road congestion calculation. To deal with the aforementioned two drawbacks and by exploiting the social feature that distinguishes normal vehicles on one hand and role and public transport vehicles on the other hand, we had the motivation to propose a pseudonym change-based scheme that uses silence periods efficiently and with safety and road congestion considerations. The proposed scheme is based on the well-known Chinese ancient war tactics plan, that is, “Give up a pawn to save a chariot” for which the exact details are given in the next sections.

### 1.5. Main Contributions

With regard to the prior works and the motivation behind this actual research, the main contributions that this work makes are presented as follows:
The proposition of a location privacy-preserving scheme (OVR) that takes into account both road safety and congestion by using Overseer vehicles (“OVR”, from where the name of the scheme was taken).The exploiting of a social feature that is the class of the vehicle: role vehicles, public transport vehicles and normal vehicles in order to achieve the main objective which is rising the overall location privacy level while maintaining a high level of road-safety and enabling road-congestion estimation and reporting.In addition to road safety and location-privacy, a Quality of Service (QoS) aspect is enhanced while using the proposed scheme. The QoS enhancement is represented by reducing the number of sent safety messages that may result in packets collusion.Enabling the feature to estimate and report road-congestion even with the present of silent vehicles and hence, the OVR scheme keeps and maintains the good functioning of road-congestion estimation.

### 1.6. Paper Organization

The rest of this research is organized as follows: In Section 2, we present the related work of the most prominent research that deals with the problematic in question. Then, the taken system model, in addition to highlighting some relevant attacks and the considered assumptions, is presented in Section 3. Next, Section 4 details and explains the proposed scheme, which is OVR (for overseers) for location privacy, road safety, and road congestion estimation. After that, the performances of OVR were put into evaluation via a set of simulations in two main scenarios—the Manhattan and the highway scenarios—and that is covered in Section 5. In Section 6, a discussion and global results interpretation is elaborated to summarize the obtained results and to discuss the most relevant facts and observations related to the OVR integration in addition to giving the limitations that can be addressed in future work. Finally, we conclude this research in Section 7.

## 2. Related Work

In this section, we discuss the prominent schemes and techniques used in recent decades to deal with the problem of location privacy, as well as congestion reporting and vehicular mobile edge computing. The incorporation of connected car technology using V2X communications [3] introduced a lot of benefits but also let researchers consider the resulting shortfalls, such as the privacy challenges for IoV users [4]. Vehicles send BSMs to achieve safety, but nonetheless, it opens up privacy issues that are mostly related to the geographic location data inside the BSMs. Pseudonymity [5] is used to substitute pseudonyms for real identities; this way, the tracker finds it more inconvenient and difficult to match the vehicle traces, especially with the use of frequent pseudonym changes [6]; however, using an intense pseudonym change frequency leads to a trade-off represented in affecting routing algorithms [7]. Changing one communication identifier is generally not enough because there are other identifiers in the same communication stack that are not changed such as mac address and hence such identifiers also have to be changed but this change has to be conducted carefully in order not to affect the other communications such as V2P and V2C.

The use of pseudonym changes is good for privacy but fails compared to some scenarios in which the tracker is more sophisticated. Such a tracker is able to make real-time vehicle traces’ tracking with the help of tracking algorithms and tools [8]. Based on this principle, a lot of schemes with enhanced tricks are used to achieve better geographic location privacy. Among the schemes, there is the mix zones concept [9] in which vehicle $v_{i}$ is mixed within a fictive zone that contains other vehicles which are similar to $v_{i}$. This group of vehicles provides good indistinguishability, yet the precise geographic location in addition to the high frequency of beaconing make it easy for the tracker to build the traces of vehicles individually. In a similar work, Beresford and Stajano [10] proposed the CMIX scheme, where they suggested the use of symmetric cryptography that relies on public keys to make sure the beaconing is encrypted and hence evades the tracking. The most relevant limitation of CMIX is the incorporation of encryption in time-critical safety applications, which is not recommended because of the involved delays similarly to the limitation coming from heavy cryptography like elliptic-curve-based and bilinear-pair-based cryptographies [11].

To cover the exploit of the pseudonym change correlation attack, Huang et al. introduced the mobile nodes’ silent period concept [12]. The silent period is a random or static period of time in which the mobile node is ceasing transmissions before emerging again with a new identifier. The random period is used to confuse the tracker when it is used in a set of other nodes, whereas the static period is used to transit between two geographic zones to cover the spatial correlation. The limitation of this approach is that we need to sacrifice safety for the sake of privacy. Based on the concept of silent period, Buttyán et al. proposed SLOW [13], a scheme that lets vehicles be silent when their speeds are below a specific threshold. The idea behind the scheme is interesting since it both confuses the tracker and ensures higher road safety because of the conditional silent period. However, not every low speed is safe, and that can be seen in the scenario of vehicles emerging from blind spots that are driving at higher speeds and hence, the fast vehicle could be unable to perceive the oncoming silent vehicle.

Focusing on places such as intersections and parking lots, Lu et al. proposed a social spots-based scheme [14]. The scheme exploits the nature of social spots to its advantage since the tracker will be more confused when vehicles proceed to change their pseudonyms in intersections and parking lots, and that is natural because of the densities inside such locations, which results in difficult tracking. Using a different approach, Pan and Li deploy the Cooperative Pseudonym Change CPN (CPN) [15] scheme that lets *k*-close vehicles make a synchronous pseudonym change in order to blur their recently updated pseudonym; nevertheless, the fine-grained geographic locations make it possible for the tracker to execute the correlation attack between the new and old pseudonyms.

Emara et al. propose a Context-Aware Privacy Scheme (CAPS) [16] and it is based on the vehicle’s context, meaning that it keeps monitoring the surrounding neighbor vehicles and when one or more other vehicles are in the silent state, the vehicle enters the silent state as well. Once its context, while being silent, becomes similar to that of the other silent vehicle(s), it leaves the silent state with a new pseudonym, which reduces tracker confusion. Their scheme is compared to the Random Silent Period (RSP) [12] scheme that is built upon entering the silent state for a random period of time, followed by a pseudonym change that results in a spatially alike mix-zone since vehicles disappear and emerge from two spatial zones. The outcome shows better traceability and QoS results for CAPS over RSP.

Exploiting the feature of transmission range adjustment, Babaghayou et al. propose Whisper [17], a location privacy scheme that lets vehicles adjust their beacons’ transmission range based on different factors like their own speed and that of the close vehicles. The authors chose to allow vehicles to shorten their transmission range at low speeds, and if other conditions are met, the vehicle proceeds into changing its pseudonym. This is why the tracker becomes confused because it is likely to lose its target’s traces when it is beaconing with a short range. The scheme was compared with SLOW, CPN and RSP and it showed both a high privacy level and provided safety since it does not cease beaconing at low speeds compared to SLOW.

The research community, governments, and automotive industry are all working on solutions to cope with the problem of road congestion. For this aim, advanced mechanisms can be used, such as the utilization of Global Positioning System (GPS) devices, cameras, and sensors, which help in giving more precise road congestion estimation as described in [18]. The use of various data sources like point sensors, point-to-point sensors, and area-wide sensors and auxiliary information for the road congestion problem can foster the precision and efficiency of the calculated road congestion estimation, as shown by Cvetek et al. [19]. The estimation of road congestion is feasible with the reliance on image-based frameworks that integrate Convolutional Neural Networks (CNN) models as conducted in the research of Gao et al. [20]. The study was performed on a large dataset of 1400 images under two weather conditions: rainy and sunny. Their scheme achieved better results compared to the other four schemes that were used in the study. With the use of a similar technique, Farooq et al. propose an image segmentation framework for road occupancy with low calculation constraints. Both the threshold segmentation and shadow removal techniques were applied during the study and have been shown to give better estimations of road occupancy. Additionally, the authors pointed out that the camera angle from where the images data are taken has an influential impact in which the closer to the road surface the centered camera is the better the results will be [21].

Transmission delays and offloading energy consumption are challenges encountered by IoV-enabled vehicles, where it is necessary to deploy a reliable scheme like that of the Joint Delay and Energy-Vehicle Computational task Offloading (JDE-VCO) scheme of Lv et al. [22]. In their scheme, the focus was on task offloading and migrations under the Software Defined Vehicular Networks (SDVN) architecture with minimum delay and energy consumption goals. Additionally, JDE-VCO focuses on data upload, and task calculation and migration. It has been evaluated and compared against a set of other optimization schemes under various vehicle densities, and their scheme showed better outcomes. Feng et al. proposed two vehicular edge computing solutions, namely (a) the Autonomous Vehicular Edge (AVE) and (b) the Hybrid Vehicular edge Cloud (HVC) [23]. AVE gives vehicles the option to operate in a decentralized environment, and thus their computing resources are shared between them when needed via offloading techniques, whereas HVC gives an extension to use other accessible entities such as RSUs and cell towers, which gives more options. Both solutions foster the effectiveness and applications range of IoVs, but many challenges remain to be addressed, such as privacy leakage while offloading [24], for global mobile edge computing adoption.

## 3. System Model and Potential Attacks

In this section, we present the fundamentals that our research was built upon. Firstly, we define the network and threat models in addition to some of the potential attacks that target geographic location privacy. Finally, the assumptions made are mentioned, and that is to define the limits of our research.

### 3.1. Description

Having different kinds of vehicles with different objectives and missions makes the location problem more complex in terms of what or who really has to be protected. Additionally, some other actors are impacting the achieved location privacy level, whether in a positive sense (from the defender side) or a negative sense (from the attacker side). In what follows, the most important actors are going to be briefly presented to give the study its general scope, in addition to the set of assumptions that were taken into account while developing the study.

### 3.2. Network Model

The network model consists of the law-side authorities, the assisting infrastructure, and the other automobile entities, as shown in Figure 3, and they can be described as follows:The law-side authorities: the different authorities needed for smoother, safer, traceable and more secure vehicular environments. Such authorities may include, but are not limited to, the Law Enforcement Authority (LEA), Certificate Authority (CA), Pseudonym Issuance Authority (PIA), Pseudonym Resolution Authority (PRA), etc. Generally speaking, these authorities have a centralized nature to efficiently manage the system in question. Many open debated question are discussed by the field expert, like “who would be trusted to control the sensitive authorities?”. The use of high performance devices such as High-Availability (HA) databases and powerful processing resources is mandatory to ensure a robust system management. The law-side authorities are considered as the cloud level which not only ensures a good network functioning, but also gives the option to reinforce the decisions and suggestions of the network vehicles with the help of the Artificial Intelligence (AI) optimization [25] by doing mass calculations at the cloud level and delivering them down to the vehicular mobile edge level [26] to act as a model.The assisting infrastructure: a medium used in the system in order to facilitate communication between the different entities. A high security level [27] has to be ensured before using this assisting infrastructure in such delicate systems to not jeopardize the safety of drivers, passengers and pedestrians. At the same time, Road-Side Units (RSUs) and cell towers have started integrating the concept of edge servers at their level [28]. The main benefit is to minimize the reliance on distant cloud data centers on the one hand and to reduce the delays for critical real-time applications [29] on the other hand.The three kinds of vehicles are as follows: normal, role and public transport vehicles all integrate mobile edge servers in their On-Board Units (OBUs) devices in order to minimize transmission delays and exploit the high processing ability provided by the integrated processors and calculation resources. They are defined as (1) normal vehicles: the principle entities that are the subject in question. Their aim is to achieve a high level of location privacy while doing their usual driving tasks. In this manuscript, they may be referred to as “privacy-seekers”, (2) role vehicles: the supporting vehicles. By nature, role vehicles have frequent tasks that depends on their specific job (police, ambulance team, fire brigade, etc.). Unlike the normal vehicles, role vehicles do not necessarily need to keep their privacy, and as a consequence, they may be utilized to boost the normal vehicles’ privacy. At the same time, they should be utilized for ensuring a better road-safety which is influenced by the normal vehicles’ behavior while seeking for higher privacy level. In this manuscript, such role vehicles may be referred to with the name as “safety-providers” and (3) Public transport vehicles: the category of vehicles that is used for public transportation. The existence of such a category is to fulfill the needs of persons’ movements. They aim at advertising their whereabouts and thus, they are not worried about their location privacy; in fact, it is the contrary. In this manuscript, such public transport vehicles may also be referred to as “safety-providers”.

The network model in IoV ought to be well-defined and contain as much mandatory entities as possible for a better and safer functioning.

### 3.3. Threat Model

In the threat model, consisting of the attacker, often called the tracker, the eavesdropping stations spread to achieve full coverage mode and many other resources are used to store and process the necessary data. The tracker’s resources are described as follows:The attacker: the principle adversary who aims to track, monitor and identify their targets based on the collected traces. Such an entity can be a criminal organization, an advertising company, a suspicious third party, etc. Additionally, the main objective of doing such a malicious job is tied to the nature of the attacker (it depends).The eavesdropping stations: a set of units that are possessed by the attacker. They are built upon the wireless technology used by vehicles (e.g., 802.11P standard) to be run over the same wireless medium which leads to collecting the sent and exchanged data between vehicles. The eavesdropping process is seen to be a preliminary step towards tracing and identifying the targeted vehicles. The collected data are sent to the end-devices to finalize the main task of the attacker.The end-devices: dedicated devices and equipment used by the attacker in order to analyze the received data that were collected from the previous step (the eavesdropping stage). Such end-devices are deployed to perform a set of operations like (a) storing the different traces, (b) building these traces and (c) matching the real identity with the pseudo-identity of the target vehicle’s traces. Thus, an important amount of storage, RAM and computational resources ought to be available for the attacker in order to perform such a malicious activity.

With these in mind, the threat model for IoV is feasible but requires a considerable amount of resources, and hence bodies with important financing are expected to play the threat model’s role.

### 3.4. Privacy-Related Attacks

IoV has a large set of security threats and attacks, but for this exact problem of location privacy [5], we highlight some of the troublesome privacy attacks that target IoV and threaten its flourishing, especially with the presence of privacy advocates who will want to make sure their privacy is not easily leaked. We recall some of the privacy-related attacks as follows:Denial of Service (DoS): for a vehicle $V_{i}$, a set of pseudonyms has to be ensured and reachable at any time $t_{i}$. However, facing a DoS attack that targets the dedicated service that provides pseudonyms would have a direct impact on the availability of these pseudonyms, and hence, $V_{i}$ will be unable to change or use new pseudonyms.Eavesdropping Attack: $V_{i}$ keeps sending a special kind of vehicles (BSM) and that is to ensure a high level of safety yet this benign feature, it also opens privacy issues exploited by eavesdropping such packets. With this definition, collecting the sent BSMs by launching the eavesdropping attack, which is preliminary to other attacks, gives the attacker a high position on monitoring, tracking and identifying their targets later.Identity Disclosure: in such an attack, the attacker reveals the identity of a neighboring vehicle $V_{j}$ using a controlled vehicle $V_{i}$. In most cases, attackers are able to use such a privacy attack after infecting $V_{i}$ with a virus [30]. $V_{j}$’s identity, as a consequence, is going to be revealed with this attack.Location Tracking: with the fine-grained locations included in BSM packets, and since the safety requirements require vehicles to periodically send such information to the neighborhood, exploiting the location field by malicious entities becomes an easy task. The obtained data are just a preamble to other exploits, like knowing the real holder of the pseudonym, their most visited places, their driving patterns, etc.Virus or Malware Attack: viruses and malwares attack computers. IoV is not a different environment for them. With this in mind, it is important to have a robust security mechanism to thwart the infection of such virus/malware attacks, especially those affecting privacy of IoV users. The type of data that can be exposed due to such an attack can be, and not limited to, the past and current traces, the most frequent places, the headings, the driver’s tendencies, etc.Man In The Middle (MITM) Attack: one of the most interesting attacks in cyber security in recent decades. In IoV, having a MITM attack means intercepting the communications that are established between two vehicles $V_{1}$ and $V_{2}$ by an attack $A_{MITM}$ and even being able to reply to each of the two vehicles without them being aware of the situation. What is next is the exploitation of critical and sensitive data that were initially meant to be between the two peers. A lot of techniques were proclaimed to be efficient, like the delay-to-respond value, which indicated an abnormal delay between the exchanged data between the two nodes (vehicles).Masquerade Attack: internal attackers are seen to be highly dangerous for cyber security systems. Masquerade attack is not that different. The meaning of this attack is to obtain an authentication mean [31], which lets the attacker’s vehicle be identified as an authenticated node in the system. Before being found out, many privacy attacks could be executed such as the extraction of some sensitive geo-location data and other attacks that would not be feasible without an authentication (or stealing an identity and its credentials). As a counter measure and mitigation, the integration of reputation-based solutions [4] is preferable.

### 3.5. Assumptions

In this research study, we put forth a set of assumptions that we consider both while developing our scheme and while simulating its performance. These assumptions are as follows:In order to be authenticated within the system, vehicles use the Public Key Infrastructure (PKI) certificate method, which is also used to establish secure communications. When a vehicle needs to change its pseudonym, a new certificate ought to be used for that.The attacker has full coverage on the monitored map zone, and hence, most packets are meant to be received by them.The geo-location data included in the BSM packets are derived from the vehicle’s GPS device.Both (a) role vehicles and (b) public transport vehicles are not concerned with preserving their privacy, yet they are concerned with ensuring high road safety.Normal vehicles can be (1) privacy-seekers to rise their privacy or (2) safety-providers to ensure high road safety.Most dangerous security threats are dealt with the use of robust security mechanisms basing on cryptography and mathematical formulas and hence, they are out of this study’s scope.

## 4. Proposed OVR Scheme

Since vehicles are diverse in their type, behavior, and missions, they provide, as a consequence, a lot of options that can be exploited depending on the appropriate domain. In the location privacy and road safety domains, the aim is to fulfill the road safety requirements without sacrificing much of the location privacy of road-goers. A lot of classifications of vehicles could take place, but in this work, we consider the three vehicle categories as explained before in the network model part. These vehicles are (a) normal vehicles, (b) role vehicles, and (c) public transport vehicles, and they will be re-mentioned in the next part.

### 4.1. Preliminary

Dealing with the trade-off between safety and privacy is constantly encountered in the majority of privacy-preserving schemes, and thus, finding a good balance between these two concepts is a key component towards an acceptable solution. In this work, we exploit one of the ancient war tactics, that is, the Chinese saying of “give up a pawn to save a chariot”. As stated earlier, a vehicle may have two states: (a) broadcasting BSMs, which lowers its privacy but raises both its safety and the other vehicles’ privacy and (b) staying silent for an amount of time, which increases its privacy but keeps the safety and the privacy of the other vehicles preserved to a certain degree, thanks to the smart management of the protocol that takes into account such a safety–privacy trade-off.

With this said, the pawn that is given refers to normal vehicles that keep broadcasting BSMs for some time in a certain degree and role vehicles that are always broadcasting BSMs in a lower degree since they do not need privacy in the first place; (in fact, it is even the contrary—they need to advertise their availability like in the case of taxis and buses). The saved chariot, as a consequence, is no more than the normal vehicles that do not broadcast BSMs, or at worst, entered broadcasting mode for few times only. Later on, the effectiveness of such a mechanism is shown in terms of the resulting outcomes.

After providing this preliminary preamble, a detailed description of the proposed scheme is given.

### 4.2. Description

The modus-operandi of OVR consists of letting normal vehicles rely on role vehicles and public transport vehicles; that are considered as overseers in order to rise their location-privacy level. Noting that in some scenarios, if role vehicles and public transport vehicles are not enough, the use of a part of normal vehicles may be needed to fulfill the location-privacy level requirement.

Vehicles adjust their broadcasting state on-the-fly based on the current condition and surrounding environment. Figure 4 shows a scenario where vehicles are spread across a road intersection while adjusting their broadcasting status to achieve a better location privacy level. Taking the same example, vehicles that are supposed to keep being on broadcast status (mentioned as “safety-providers”) do formulate safety zones. A safety zone, as a consequence, is composed of at least two safety-providers and may include a couple of vehicles on radio-silence status (referred to as “privacy-seekers”). If the zone is not covered with the natural safety-providers, some privacy-seekers will enter into the broadcasting status and become temporal safety-providers and that is to fulfill the global location-privacy level requirement.

Practically speaking, this configuration would result in having at least two safety-providers that are going to be considered and called “delimiters” (a front and a rear delimiter) for the privacy-seekers inside the zone. This formulation is going to be established back and forth as long as vehicles are rolling on.

The high-tier location privacy level is not the only big benefit from using the OVR scheme, but another interesting feature can be exploited: the congestion estimation even when a lot of vehicles are already in the invisible state (the “privacy-seekers” state). It is thanks to the delimiters and the other vehicles that are in the zone. When the speed is low and there are delimiters across the road, a vehicle $v_{i}$ can estimate the density on its road or lane and, hence, can give an approximate congestion value based on these parameters. $v_{i}$ uses its integrated edge computer to calculate the estimated congestion and send it up to the edge server located at the cell tower or the RSU level [32]. By doing so, congestion-aware applications [33] can work without experiencing potential delays that may occur in the conventional cloud computing architecture. Only a portion of the generated data ought to be sent to the cloud. Figure 5 shows the general idea of how a vehicle $v_{i}$ estimates the congestion distance present when a congestion situation is happening.

Let Vehicle_l, Gap_d and Congestion_e be the vehicle length, the gap distance, and the estimated congestion, respectively. Additionally, let Distance($v_{i}$,$v_{j}$) be the congestion distance Congestion_d between vehicle $v_{i}$’s and $v_{j}$’s front edges (calculated using the sent safety-broadcasts). Congestion_e can be calculated as follows:(1)Congestion_e≈Congestion_d/(Vehicle_l+Gap_d)+1

Equation (Equation 1), in this context, does also indicate the approximate number of vehicles in a dense lane.

Having frontal congestion is just one scenario; in reality, different congestion scenarios may happen. We take the example of a one-way road with two lanes. In this example, four different scenarios may happen, and the congestion estimation may be reported to the edge server up to four times (direct front, direct rear, adjacent-front and adjacent-rear) and when reported this way, the central authority responsible for the road management will have a fair indication of the congestion situation happening on that map spot after this information is transmitted up to the central cloud [34]. All of the four scenarios are illustrated in Figure 6.

### 4.3. Advantageous Features and Potential Drawbacks

Using a mechanism such as OVR is something special since vehicles do behave differently according to the kind of the vehicle and to the current situation that either forces vehicles to start broadcasting safety messages or lets them stay in the silence mode. All of this leads to (1) advantageous features and (2) some potential drawbacks and limitations. We state the most prominent ones for the first category as follows:

#### 4.3.1. Advantageous Features

Exploiting the nature of role vehicles and public transport vehicles in favor of the overall location-privacy level.On-the-fly state adaptation for normal vehicles where they shift between broadcasting (safety-mode) and keeping the silence mode (privacy-mode) with regard to the overall road-safety.A packets-congestion-aware scheme where silent vehicles contribute to reducing the network overload and channel saturation that may occur as a result of the intense safety-packets broadcasting.A road-congestion-aware scheme where even having silent vehicles on the road does not affect the road-congestion evaluations by the different present vehicles.High compatibility with the edge computing paradigm and mainstream drivers where congestion applications have a delay-sensitive functioning that is not tolerated in some scenarios.A very acceptable trade-off between privacy and safety, thanks to the overseers that maintain both of the location-privacy and the road-safety of the whole vehicular environment.

#### 4.3.2. Potential Drawbacks

When the number of role vehicles and public transport vehicles are not enough to satisfy the OVR scheme’s requirements, a part of the normal vehicles that are chosen according to the dedicated protocol (detailed later) would sacrifice their own privacy for the sake of the other vehicles’ privacy for an amount of time.Some road maps and environments may not be suitable for the OVR scheme’s techniques, which would render the scheme ineffective and make it useless.The scheme needs its approval and implementation in most vehicles with their respective types. If only a few vehicles are applying it, then the overall results are not going to be promising.Vehicle lengths may affect the road-congestion estimations and reporting to some degree, especially since knowing only the distance between the delimiters does not give a certain knowledge about the exact number of vehicles in between.

### 4.4. OVR Protocols and Algorithms

OVR uses different protocols to enhance the experienced location-privacy but in respect to road-safety as a primary goal, as mentioned previously. The option to keep an eye on traffic congestion is also possible, thanks to the estimating and reporting protocols that work in conjunction with the edge computing paradigm for real-time perception. In this part, we detail the modus-operandi of such protocols, most of which are implemented to work after the BSM beacon reception event. However, before diving into them, we first define the various local variables used by the OVR scheme for both privacy-seeker and safety-provider vehicle classes, which are presented in Table 1 for double type, Table 2 for integer and Table 3 for boolean variables:

Some other terminologies with other types were used as follows:My_set: of a set type and it represents the set of monitored or encountered vehicles by $v_{i}$ in the current moment *t* (it has “vehicle” as its unit).My_init_set: of a set type and it represents the set of the encountered vehicles by $v_{i}$ right before entering into the silence mode; i.e., it is a snapshot of My_set in the moment of entering into the silence mode (it has “vehicle” as its unit).My_final_set: of a set type and it represents the final set of vehicles that were encountered by $v_{i}$ while in the silence mode. In other words, My_final_set is the intersection between My_set and My_init_set (it has “vehicle” as its unit).bcn: of a msg type and it represents the safety beacon message used by vehicles for safety purposes (it has “wave_message” as its unit).

Now that the different variables and terminologies used in OVR are presented, we provide in a kind of state diagram the lifecycle of vehicles that are existing in the OVR scheme as in Figure 7. In a nutshell, whenever the vehicle is turned on, it starts monitoring the surrounding environment and keeps updating its local variables, and depending on the set parameters, it takes decisions during the other stages of its lifecycle. For instance, whenever it receives a bcn message, it updates all its local variables, and right after spending the Beacon_time_cycle, it moves into three main decision-making processes: (1) change the pseudonym or not? (2) report congestion or not? and (3) send a BSM or not? in addition to re-initializing the local variables in the event of returning to the monitoring state. The behavior of OVR is also presented in a kind of four pseudo-algorithms that are intended to show the protocols’ functioning as separated modules and are divided into the following:BSM reception.Timer expiry.Pseudonym changing.Congestion reporting.BSM sending.

These modules’ pseudo-algorithms are explained and shown as follows:

#### 4.4.1. BSM Reception

When a vehicle $v_{i}$ receives a BSM packet from $v_{j}$, $v_{i}$ increments, the neighbors counter n_Neighbor first then it calculates the distance between the two vehicles and according to the distance, it may add $v_{j}$ to the set My_set. Additionally, the counter n_Neighbor is going to be a parameter for $v_{i}$ to decide later whether to be in the silence mode (being a privacy-seeker) or the broadcasting mode (being a safety-achiever). In a next step, if both $v_{i}$ and $v_{j}$ are on the same road and have the same direction, this leads one to consider $v_{j}$ as one of the two front delimiters or two rear delimiters according to the position of $v_{j}$ with regard to $v_{i}$’s location. A continuous check is performed to update the delimiter in question (the four delimiters) by searching for the furthest delimiter. The check aims at obtaining fewer delimiters for higher privacy on the one hand and covering wider map regions on the other hand. At last, the On_bcn_Reception procedures calls for another procedure which is Process_Beacon(BSM). The processing of the received beacon that is going to be conducted by other upper layers and applications is not considered in this work and hence is out of our scope. Such details are presented in the Pseudo-Algorithm 1.

#### 4.4.2. Timer Expiry

After the reception of the different beacons produced by $v_{i}$’s neighboring vehicles, $v_{i}$ would come to the time when it needs to attempt to broadcast its safety beacon; that is, when its timer expires after the beacon time slot is reached. Before sending its beacon, $v_{i}$ checks if the pseudonym change is necessary and worthy or not; this is in the Should_I_Change_my_Pseudonym procedure. A next check is for the congestion reporting option provided by the Should_I_Report_the_Congestion procedure. A last check is whether to send the generated beacon or not, and that is in the Should_I_Send_a_BSM procedure. The three checking procedures are explained in the what is presented below in more detail. After the expiry of the timer, a re-initialization of the different variables is performed preparing for the next beacon’s reception time cycle; this is ensured by the Re−initialize(). Pseudo-Algorithm 2 shows the timer expiry event.
**Algorithm 1** bcn Reception1:**procedure**On_bcn_Reception(beacon* bsm)                                            ▹ for each vehicle $v_{i}$ receiving a beacon bsm from $v_{j}$2:    Calculate the temporary distance Tempo_dist between $v_{i}$ and $v_{j}$;3:    **if** (Tempo_dist<=Delimiter_radius)
**then**                                                   ▹ if $v_{j}$ is a potential delimiter4:        **if** (Tempo_dist<=Prv_Neighbor_radius)
**then**                                       ▹ if $v_{j}$ is in the privacy neighbors radius5:           $My\_set.insert(v_{j})$;                                                                    ▹ insert $v_{j}$ in My_set6:        **end if**7:        **if** ($v_{i}$ and $v_{j}$ are on the same road) **then**8:           **if** (($v_{i}$ and $v_{j}$ are on the same direction) **AND** ($v_{j}$ is in front of $v_{i}$)) **then**9:               **if** (($v_{i}$ and $v_{j}$ are on the same lane) **AND** (Tempo_dist is > than Front_1_dist)) **then**        ▹ trying to obtain the farthest same lane front delimiter10:                   Front_1_id←bsm.SenderID();                                                           ▹ which is $v_{j}$11:                   Front_1_dist←Tempo_dist;                                                          ▹ updating Front_1_dist12:                   Front_1_speed←bsm.SenderSpeed();                                                  ▹ updating Front_1_speed13:               **else if** (Tempo_dist is > than Front_2_dist) **then**                                       ▹$v_{i}$ and $v_{j}$ are not on the same lane14:                   Front_2_id←bsm.SenderID();                                                        ▹ which is $v_{j}$15:                   Front_2_dist←Tempo_dist;                                                    ▹ updating Front_2_dist16:                   Front_2_speed←bsm.SenderSpeed();                                               ▹ updating Front_2_speed17:               **end if**18:           **else if** (($v_{i}$ and $v_{j}$ are on the same direction) **AND** ($v_{j}$ is behind of $v_{i}$)) **then**19:               **if** (($v_{i}$ and $v_{j}$ are on the same lane) **AND** (Tempo_dist is > than Rear_1_dist)) **then**          ▹ trying to get the farthest same lane rear delimiter20:                   Rear_1_id←bsm.SenderID();                                                          ▹ which is $v_{j}$21:                   Rear_1_dist←Tempo_dist;                                                      ▹ updating Rear_1_dist22:                   Rear_1_speed←bsm.SenderSpeed();                                             ▹ updating Rear_1_speed23:               **else if** (Tempo_dist is > than Rear_2_dist) **then**                                      ▹$v_{i}$ and $v_{j}$ are not on the same lane24:                   Rear_2_id←bsm.SenderID();                                                       ▹ which is $v_{j}$25:                   Rear_2_dist←Tempo_dist;                                                   ▹ updating Rear_2_dist26:                   Rear_2_speed←bsm.SenderSpeed();                                             ▹ updating Rear_2_speed27:               **end if**28:           **end if**29:        **end if**30:    **end if**31:    n_Neighbor++;                                                     ▹ increment the n_Neighbor number of $v_{i}$32:    Process_Beacon(BSM);33:**end procedure**

**Algorithm 2** Timer Expiry
1:
**procedure**
On_Beacon_Time_Cycle_Expiry()
    Should_I_Change_my_Pseudonym();    Should_I_Report_the_Congestion();    Should_I_Send_a_BSM();    Re-initialize();       ▹ re-initialize the variables preparing for the next timeslot2:
**end procedure**



#### 4.4.3. Pseudonym Changing

Each vehicle in the system must change its pseudonym every two minutes; however, for a better exploiting of our scheme and to confuse the attacker, checking if the pseudonym change is beneficial or not is crucial in this stage. This is ensured by the Should_I_Change_my_Pseudonym procedure. When a $v_{i}$ is in the privacy-seeker mode, emerging again with a new pseudonym makes the tracker confused since matching the changed pseudonym would be difficult both spatially and temporary. Changing the pseudonym is based on some conditions, which are as follows:the speed variation, due to having a regular speed between being silent and broadcasting, means that the next spot from where $v_{i}$ emerges would be easy to guess, and hence, a useless pseudonym change if performed.if $v_{i}$ is only silent for a very short period of time, changing its pseudonym and appearing again from a near spot would be easily detectable by the tracker.if $v_{i}$ keeps in the silence mode for very long period of time; this brings greediness in the vehicular network, hence forcing it to leave that state in order to perform an effective pseudonym change. This also ensures that the other vehicles would have a fair chance to be privacy-seekers and to enhance their privacy experience on one hand and to fulfill the system vivacity and assiduity requirements on the other hand.for the distance, a long distance is preferred for pseudonym change but, a very short one means a higher chance of linking the newly changed pseudonym and that is the reason for only checking the minimum crossed distance before proceeding into changing the pseudonym.a minimum number of neighboring vehicles that are stored in the set My_set is also required to be greater than or equal to the k_prv threshold. The aim here is for $v_{i}$ to be mixed in a large set of encountered vehicles and that it is seen as similar to $v_{i}$ in the eyes of the tracker.the last condition is Set_novelty_ratio that is responsible for evaluating the recency of the set that $v_{i}$ is keeping and updating. Having a lot of new vehicles entering the monitored vehicles set My_set and comparing it with the initial set My_init_set must have an acceptable ratio in order to ensure a certain level of tracker confusion.

These detailed conditions are shown in the below Pseudo-Algorithm 3.
**Algorithm 3** Change the Pseudonym1:**procedure**Should_I_Change_my_Pseudonym()2:    **if** ((the speed variation, i.e., the difference between Max_speed and Min_speed), is ≥Speed_var_thr) **AND** (the time while $v_{i}$ was silent is ≥Min_silent_time) **AND** (the time while $v_{i}$ was silent is ≤Max_silent_time) **AND** (the distance while $v_{i}$ was silent is ≥Min_silent_dist) **AND** (the set My_set of $v_{i}$ is ≥ the threshold k_prv) **AND** (the Set_novelty_ratio value is ≥10%)) **then**3:           Change_the_Pseudonym();                                      ▹ and reinitialize the regular pseudonym change timer4:    **else if** (the time while $v_{i}$ was silent is >Max_silent_time) **then**            Force v_i to leave the silence state in order to ensure efficient pseudonym changes and achieve the vivacity and assiduity requirements.5:    **end if**6:**end procedure**

#### 4.4.4. Congestion Reporting

Since staying silent lets the network vehicles be transparent to most of the safety beacon-based applications, including the congestion-aware applications, we provide a mechanism to estimate the existing road congestion while using the OVR protocol. The mechanism is triggered by the timer expiry first then if a set of conditions is met $v_{i}$, which is in the broadcasting mode, proceeds into the road congestion reporting. The set of conditions is (1) the speed of $v_{i}$ when it drops below a certain threshold, that is, Eps, then $v_{i}$ monitors all of its two front and two rear delimiters and checks for their individual speeds, when below the same defined threshold Eps, $v_{i}$ calculates the estimated congestion between itself and the delimiter $v_{j}$. This is going to be using the already-presented formulation in the OVR description part (i.e., Equation (Equation 1)). After that, the Report_Congestion procedure is executed and can be up to four times, which is the number of possible delimiters. The Report_Congestion procedure is shown below in Pseudo-Algorithm 4.
**Algorithm 4** Report Congestion1:**procedure**Should_I_Report_the_Congestion()2:    **if** (the speed of $v_{i}$ is ≤Eps) **then**3:           **for each** existing Delimiter_i
**in** the front or rear delimiters of $v_{i}$ **do**4:                   **if** (the speed of Delimiter_i is ≤Eps) **then**                        Congestion_e(distance between $v_{i}$ and Delimiter_i);                        Report_Congestion();5:                    **end if**6:           **end for each**7:    **end if**8:**end procedure**

#### 4.4.5. BSM Sending

The last procedure is SendaBSM; it occurs right after the timer expiry. First of all, if $v_{i}$ is from the safety-provider class it just sends the BSM message right away. If $v_{i}$ is from the privacy-seeker class, it looks for the condition of sending or just keeping silent. The condition is satisfied only when the number of neighbors n_Neighbor is above the k_sft threshold, in addition to the presence of at least two delimiters that are not of the same type, meaning one front and one rear delimiter. This later requirement aims at ensuring the map coverage and the presence of dispersed broadcasting vehicles with a minimum number of broadcasters that, at the same time, achieve a fair safety level while providing a high privacy level for the vehicles of the privacy-seeker class. While staying silence, $v_{i}$ calculates locally its Set_novelty_ratio value. This later is based on My_init_set, which is the first set of neighbors of $v_{i}$ before entering silence, and it is compared with the final set My_final_set which is the newly encountered vehicles by $v_{i}$ while being silent, in addition to taking My_set during the process since it gives the set of vehicles in the instant *t*. The value of Set_novelty_ratio is needed in the step of checking the eligibility of changing the pseudonym which was discussed earlier in Pseudo-Algorithm 3. As a last step, a quick initialization is performed and is represented by clearing the two sets My_set and My_final_set. Pseudo-Algorithm 5 describes the procedure of SendingaBSM.
**Algorithm 5** Send a BSM1:**procedure**Should_I_Send_a_BSM()2:    **if** ($v_{i}$ has a Car_type of “role vehicles” or “public transport vehicles”) **then**3:            Send_the_BSM_Message();                   ▹ these two kinds of vehicles always broadcast4:    **else if** ((the number n_Neighbor is ≥k_sft) **AND** (Front_1
**OR**
Front_2) **AND** (Rear_1
**OR**
Rear_2)) **then**5:            Delete_the_BSM_Message();6:            **if** (My_set.size() == 0) **then**7:                    Set_novelty_ratio=0;8:            **else**9:                    Set_novelty_ratio= 1 - My_final_set.size() / My_set.size();10:            **end if**11:    **else**12:            Send_the_BSM_Message();13:    **end if**14:    My_set.clear();                              ▹ a reinitialization for the next timeslot15:    My_final_set.clear();                          ▹ a reinitialization for the next timeslot16:**end procedure**

## 5. Performance Evaluation

In order to evaluate the effectiveness of OVR, a set of simulation runs were taken for two different scenarios: a Manhattan grid scenario and a highway scenario.

Before going into the performance evaluation, the set of all used evaluation metrics in this work is explained in the following part.

### 5.1. Evaluation Metrics

To quantify the level of achieved privacy or the enhanced Quality of Service (QoS) of a specific scheme, a lot of privacy [5,35] and QoS [36] metrics were proposed in the literature. In this paper, we focus on some of these evaluation metrics and consider them while evaluating our scheme versus other schemes, more precisely: CAPS [16], CPN [15], RSP [12] and SLOW [13] (their mechanisms were already detailed in the related works section). The first category of metrics is based on the privacy concept, and we mention them as follows (they are mentioned in [35] as well):Traceability: it is a location privacy metric that represents the correctness of reconstructing the vehicle traces from the generated beacons that are obtained by the adversary.Normalized traceability: the traceability metric takes all vehicles into consideration while building their traces but sometimes not all of them change their pseudonym. Building their traces becomes something evident, and thus excluding such vehicles gives more meaning to the metric; only consider vehicles that did change their pseudonym.Maximum anonymity set size per trace: the anonymity set size is defined as the indistinguishability between the vehicle $v_{i}$ and the other vehicles that are in the vicinity. Now the maximum anonymity set size per trace is the maximum achieved anonymity by $v_{i}$ during its driving experience.Maximum entropy per trace: the anonymity set size takes all vehicles as equal while in fact some vehicles are more likely to be similar to the monitored vehicle $v_{i}$. With this said, the entropy is defined as the uncertainty in a random variable [37]; in other words, the entropy is the measure of a vehicle’s anonymity inside the particular set of vehicles where not all these vehicles are alike with the monitored vehicle $v_{i}$. The equation of the entropy is given in Equation (Equation 2) as follows:
(2)$$H_{p}=-\sum_{i=1}^{\left|AS\right|}p_{i}\log_{2}p_{i}$$Noting that |*AS*| is the set of anonymity size, *p_i_* is the probability of being $v_{i}$ as the target. The more the probabilities of vehicles being the monitored vehicle $v_{i}$ are equal, the more their entropy will be and the higher it will be (Hmax). This highest value is given in Equation (Equation 3) as follows:
(3)$$\forall i:p_{i}=\frac{1}{\left|AS\right|},H_{pmax}=-\sum_{i=1}^{\left|AS\right|}p_{i}\log_{2}p_{i}=\log_{2}\left|AS\right|$$Now the maximum entropy per trace is the maximum achieved entropy by $v_{i}$ during its driving experience.

The second category of metrics is based on the QoS concept, and we mention them as follows:Total pseudonym changes: it represents the sum of all pseudonym changes conducted by the set of available vehicles during the simulation time. In other words, it is the sum of regular pseudonym changes and special pseudonym changes. The higher it is, the better the location privacy will be, but at the same time, the worse the network communications and the pseudonym pool refilling will be, and vice versa in the case of lower pseudonym changes; hence, finding a balance between them will give a better trade-off.Total sent beacons number: it simply represents the total number of beacon messages broadcast by the set of available vehicles during the simulation time. The higher it is, the better the safety level will be, but at the same time, the worse the network communications, overhead and packet congestion will be.Total verified signatures: it represents the total number of verified beacon messages by the receiver. Its utility comes mainly from the data integrity validation, where only authorized and authenticated vehicles are able to participate and send such cooperative beacons. Even though current compute resources are from very good, receiving so many beacons and the requirement to verify them in a short period of time would be a big challenge.

Another category of metrics is taken for our OVR scheme, which is related to the road congestion mechanism and is defined as follows:Reporting congestion vehicles number: it represents the number of vehicles that did report the congestion at least once. This gives an indication of the vehicles that experienced congestion on the road during their driving journey.Road congestion reporting times: it represents the total number of times a road congestion was reported by all existing vehicles.Road congestion value: it represents the total road congestion value for all existing vehicles. This specific value is built relative to the OVR scheme and hence is customized. Each vehicle can report up to four values per Report_Congestion event.

### 5.2. Simulation Setup

As a means to validate our proposed OVR scheme, we chose the simulation method. Two scenarios were taken: (a) a Manhattan grid scenario and (b) a highway scenario. Figure 8 illustrates the setup in a graphical view, while Table 4 describes in more detail the taken configuration in terms of mobility, environment, evaluation and strategy. Some other strategy configurations were taken as their default values from PREXT configuration [35] (described next).

With the setup of the Manhattan and highway scenarios in addition to the simulation parameters in place, studying the behavior of each scheme and comparing it between them becomes feasible. The first evaluation is for the achieved location-privacy level; next comes the achieved QoS and finally the congestion reporting functionality is evaluated, which is applicable only in the more dense environment, i.e., the Manhattan scenario.

Concerning the validation, we used a set of simulation tools and frameworks to evaluate the effectiveness of the proposed OVR scheme. To establish the Manhattan grid and the highway scenarios, we used the NETEDIT mini-tool included in SUMO [38], which is considered credible and one of the mobility simulators that are considered to be realistic. As for the network simulator, the choice was on using OMNeT++ [39] that is a component c++ based simulator that uses discrete events. OMNeT++ is so flexible that it can run many frameworks depending on the use case; in our scenario, we used Veins [40], the vehicular network simulator, which acts as a bridge between both (a) the mobility simulator SUMO and (b) the network simulator OMNeT++. Lastly, for the location privacy evaluation, we used the extension PREXT [35], which is a framework developed based on Veins codes and methods. PREXT was developed by Emmara et al. and presents (1) a set of location privacy schemes ready to be used, (2) a set of location privacy metrics and (3) a set of QoS metrics. Similarly to the schemes provided in PREXT, we created our location privacy scheme OVR with the aim of comparing its performances against some of the other schemes that are already in PREXT. It is also important to know that another layer was added to OVR that is related to road congestion detection and reporting, in addition to creating customized road congestion metrics to quantify the existing road congestion when it happens.

### 5.3. Privacy Simulation Results

For the privacy simulation results, the two scenarios are evaluated using the (1) Traceability, (2) Normalized-traceability, (3) Maximum anonymity set size per trace and (4) the Maximum entropy per trace metrics.

#### 5.3.1. Manhattan Scenario

The Manhattan scenario is known to have more intersections, traffic lights, unpredictable vehicle maneuvers and is a grid-based map. All of these characteristics have different influences on the privacy schemes. The results of the location-privacy evaluation are given as follows:

##### The Achieved Traceability

Using the traceability metric and taking into account the obtained results in Figure 9, the two schemes, namely CAPS and CPN, did not perform well, especially in the case of CPN. The reason behind that is that CPN relies on neighboring vehicles to perform the pseudonym change. The different vehicle densities in this map scenario were not enough to give better privacy results for the two aforementioned schemes, with a slightly better performance when the density reached 200 in the case of CAPS. RSP started with about 80% of traceability and it was enhanced across the higher densities, where it reached below 50% in the density of 200. As for SLOW and OVR, SLOW was not traceable by the adversary, which is a good indicator, but this high privacy value was caused by the relatively low vehicle densities in such a map scenario. Talking about OVR, the traceability value even started with a promising percentage, which is 5%, and kept improving across the higher densities, so in general, we say that OVR had outperformed the other schemes with the exception of SLOW where the reason is the nature of the map scenario and the silence at low speeds known from SLOW protocols.

##### The Achieved Normalized-Traceability

With the same logic as the previously obtained results, we obtained almost the same behavior from the five schemes as in Figure 10 with an exception now after excluding those vehicles that did not change their pseudonyms during the tracking performed by the adversary. CPN, RSP and SLOW achieved almost the same privacy level across the simulation runs in the different densities. However, CAPS performed well and that is apparent in the high densities, namely in 150 and 200 vehicles densities where it achieved 76% and 71%, respectively. As for OVR, the obtained results were more promising even than those once obtained in the regular traceability metric, i.e., lower than 3% in all densities.

##### The Achieved Maximum Anonymity Set Size per Trace

In the maximum anonymity set size per trace evaluation, the first observation that we have after seeing Figure 11 is that the more we increase the vehicle density, the more the obtained privacy level is enhanced, where the higher it is, the better it will be. CPN now outperforms the other schemes (giving a maximum anonymity set size ranging between 6 and 10), mainly because it benefits from the close, crowded neighbors for each vehicle, but it cannot be taken as a proof since this metric alone is not ensuring a high level of privacy in all cases. As for the remaining schemes, they almost performed the same and did the same convergence in high densities, with remarkable low privacy values for CAPS and RSP (below the value 2) and better results for our OVR scheme plus SLOW (at about the value 3). This conclusion puts OVR and SLOW among the best privacy insurers.

##### The Achieved Maximum Entropy per Trace

For the last privacy evaluation, which is dedicated to the maximum entropy per trace shown in Figure 12, almost the same behavior was observed on the five schemes except that the OVR scheme showed a clearer outperformance (of about 0.85 at the 200 density) against the remaining schemes without counting CPN. This indicates that in higher densities, the OVR scheme shows good privacy potential. This is a good sign for the more crowded map scenarios.

#### 5.3.2. Highway Scenario

The highway scenario is known for its monotonous driving over long distances with few speed variations, as well as other specific characteristics that influence privacy schemes, either positively or negatively, depending on the specific privacy scheme. The results of the location-privacy evaluation are given as follows:

##### The Achieved Traceability

Unlike the previous Manhattan scenario, the traceability performance of the five schemes under the highway scenario shown in Figure 13 gave different results. Except for the OVR scheme, the remaining privacy schemes produced poor privacy levels, represented in almost full traceability values (of about 95% to 100%) by the attacker. The interpretations would show that for SLOW, for example, a monotonous driving pattern during the whole driving in such a scenario leads to very rare slow speed occasions which has a negative effect over the achieved location privacy level. The same logic applies to the schemes that rely on the neighboring vehicles, i.e., cooperative-based schemes, and that describes the poor results that are noted, which are due to the disparity and high distances between vehicles in highway scenarios. In what concerns the OVR scheme, it exploited the nature of the overseers, or in other words, the delimiters. Driving long distances leads to dynamic formulations that benefit the privacy-seekers by (1) enhancing their privacy level and (2) still fulfilling the safety requirements that are achieved by the safety-providers. Additionally, a small privacy degradation is observed when the vehicle densities become augmented (go below 60%).

##### The Achieved Normalized-Traceability

In the normalized traceability performances shown in Figure 14, which excludes the vehicles that did not change their pseudonym, our OVR scheme keeps on outperforming the remaining schemes with a remarkable enhancement of about 10%. Another observation is for both SLOW and CAPS, because there were no available occasions to perform the pseudonym change at low speeds for SLOW and no encountered context to do the pseudonym change for CAPS led to null location privacy, which means that the evaluation was not possible under this scenario and these parameters. CPN and RSP keep the same results as those earlier in the regular traceability evaluation (of about 95% to 100%).

##### The Achieved Maximum Anonymity Set Size per Trace

Differently from the two previously metrics, Figure 15 shows that both SLOW and CAPS achieved no maximum anonymity set size per trace (the 0 value), and this is due to the same reasoning, which is the lack of unpredictable movements since all vehicles are driving in a monotonous way. On the other hand, RSP and OVR were at a low privacy level compared to their performances in the Manhattan scenario (now they achieve 0.8 and 1.2, respectively). As for CPN, it started with a low privacy value (0.3) but improved over high densities (2.8 in the highest density), yet, the maximum achieved value was still below that of the Manhattan scenario, mainly because of the mobility patterns in the highway scenario.

##### The Achieved Maximum Entropy per Trace

Lastly, for the achieved maximum entropy per trace for the highway scenario that is shown in Figure 16, we remark that all of the schemes started with a low location privacy value (in between 0 and 0.09) but with the increase in vehicle densities, we observe three different curve developments: (a) SLOW and RSP improve only slightly with the increase of vehicle densities (reached in between 0 and 0.02), (b) our OVR scheme achieved a visible high location privacy level (of about 0.17) and (c) CPN achieved the highest improvement among the five schemes (a value of 5.2) which was expected due to the nature of the scheme that is based on the number of close neighbors, which indeed becomes higher with higher vehicle densities.

### 5.4. QoS Simulation Results

The QoS metrics are mainly used to evaluate the network performances and the overhead costs. For their simulation results, the two scenarios are evaluated using the number of (1) consumed pseudonyms, (2) sent beacons and (3) verified beacon signatures.

#### 5.4.1. Manhattan Scenario

The Manhattan model influences the QoS in different ways, and that is foreseeable because of the nature of the crowded roads and intersections, the fast topology changes and the presence of buildings and obstacles. The obtained QoS results are presented as follows:

##### The Number of Consumed Pseudonyms

Each scheme has its own way of dealing with the pseudonym change and expiry policies. Figure 17 shows the total number of pseudonym change with the increase in vehicle densities for the four schemes where it is apparent that CAPS had the lowest pseudonym changes number for all densities (which reached 200 at best) followed by the three schemes, namely RSP, SLOW and OVR, which were all in almost the same band (in between 550 and 800) with OVR being on the top of them for higher pseudonyms consumption. While the remaining scheme, which did consume a huge number of pseudonyms, was CPN (augmenting from 6 K to 35 K). The reason behind it is the nature of CPN which exploits the nature of close neighbors to perform the pseudonym changes to confuse the attacker, and since we used the default value of the *k* neighbors as 2, this resulted in such a high pseudonym changing rate for all densities.

##### The Number of Sent Beacons

The number of generated beacons is essential for the safety applications and is determined by the used privacy scheme. Figure 18 shows the number of generated and sent beacons with the variation of vehicle densities in which CPN is the scheme with the most sent beacons (up to 42 K) and this is due to the scheme’s nature that does not use silent periods, followed by CAPS (reaches 40 K) then RSP (reaches 37.5 K) and all of them are considered to be high beacon senders. After that comes OVR which ends up at a lower value in the density of 200 vehicles (gives about 29 K) compared to the first three schemes. At last, SLOW achieves the lowest value (20 K at best) of sent beacons in all vehicle densities. This is due to the silent periods involved in the scheme when the speeds of vehicles are low.

##### The Number of Verified Beacon Signatures

By receiving a beacon message, a verification process of the beacon’s signature is triggered locally at the receiver’s level. Figure 19 shows the growth of the verified signatures with the increase in vehicle densities for the five schemes. The first observation is that the five schemes make a quadratic-like evolution and are almost similar to the analysis of the sent beacons figure (relative to the sent beacons); CPN, CAPS and RSP experience the highest signature verification number (1.9 M, 1.87 M and 1.72 M, respectively) followed by OVR with moderate values (reaches 1.28 M) and in last place comes SLOW (890 K) which achieves low values due to the proportional relationship between the sent beacons and the respective verified beacon signatures.

#### 5.4.2. Highway Scenario

The highway model influences the QoS in different ways as well because of the way vehicles move in the long distances with long road segments which makes the topology less frequently-changing and hence no such routing table changes. Additionally, the density of vehicles becomes low, and hence packet congestion becomes rare, especially with the lack of surrounding buildings and obstacles; it all comes in favor of QoS applications. The obtained QoS results are presented as follows:

##### The Number of Consumed Pseudonyms

In the highway scenario, the five schemes always base their behavior on the pseudonym change concept to preserve the location privacy of the vehicles, but due to the nature of the map, the behavior changes. Figure 20 shows almost the same pseudonym change development as that of the Manhattan scenario for the three schemes, namely CPN, RSP and OVR. Now for the exception, both CAPS and SLOW did not make any triggered pseudonym changes for all densities, and this is due to the monotonous and continuous nature of the long highway road, which prevented the two schemes from performing useless and predictable pseudonym changes when the speeds of vehicles were too high and where no appropriate context was available for SLOW and CAPS, respectively.

##### The Number of Sent Beacons

Contrary to the Manhattan scenario, where vehicles are on a square-like map while generating and sending their beacons, in the highway scenario, vehicles are lined up in a straightforward manner due to the highway nature, and this impacts the behavior of the privacy schemes, especially those with neighbors and topology dependencies. Figure 21 shows that CAPS, CPN and SLOW are identical in terms of the generated beacons (about 15 K), then comes RSP with lower values (about 13.2 K); this is due to the random silent periods related to the RSP scheme which prevented sending some beacons. At last, OVR has the lowest values (about 6.4 K); this is due to the scheme’s ad hoc nature that lets few vehicles be safety-achievers, which lets the other ones be privacy-seekers, hence reducing the sent beacons while maintaining a good safety level.

##### The Number of Verified Beacon Signatures

Following up the last point, that is, the sent beacon number analysis, the impact on the received beacon messages is much more apparent in the highway model since a one beacon broadcast is likely to reach fewer vehicles than that of the Manhattan scenario due to the same reason, i.e., the highway line-like nature, but this does not affect the fact that each beacon broadcast results in many beacon receptions, which leads to receptions close to the quadratic-like evolution. Figure 22 shows that all of CAPS, CPN and SLOW achieved the highest verified beacon signatures (of about 144 K) followed by RSP (about 130 K) for the same reason discussed in the previously stated point. At last, OVR shows the lowest value of verified beacon signatures (about 62 K) due to the same reason discussed earlier as well.

### 5.5. Road Congestion Reporting Results

As for the road congestion reporting simulation results, only the Manhattan grid scenario model was used, and due to the nature of this model, a couple of road congestion events are encountered by an important number of vehicles, which is contrary to the highway scenario where the topology of the network does not change so frequently and the vehicles drive monotonously, so road congestion is likely not going to happen and hence, we exclude its evaluation in the highway scenario. In this evaluation, three metrics were taken: the number of (1) reporting vehicles, (2) road congestion reporting and (3) the total road congestion value.

#### 5.5.1. The Number of Reporting Vehicles

When experiencing a road congestion, $v_{i}$ proceeds into reporting this event with the aim of informing the managing authorities at first and to make the other vehicles aware about the existing road congestion later on by spreading the congestion information from the preprocessed data at the managing authorities level. Figure 23 shows the number of reporting vehicles in the different vehicle densities while using the OVR scheme. In the Manhattan scenario, most of the road congestion events are generated at the intersection level, and the more vehicles that are present, the more road congestion that is experienced and reported. At the first density, fewer than 40 vehicles among 50 did report road congestion at least once. With the advance of vehicle densities, the number of vehicles that report road congestion comes close to the total number of vehicles per density as follows: about 90, 145 and 199 which match 100, 150 and 200, respectively, and this indicates the convergence of congestion reporting vehicles towards the total number of existing vehicles.

#### 5.5.2. The Number of Road Congestion Reporting

Moving now to the total number of road congestion reporting along all the simulation duration in OVR, Figure 24 shows that it begins with about 1.5 K reporting times at the 50 vehicles density, and keeps increasing in a linear-alike way, i.e., almost 4.5 K, 8.5 K and 12 K, which match 100, 150 and 200 of vehicle densities, respectively.

#### 5.5.3. The Total Road Congestion Value

The last congestion reporting metric is the total road congestion value reported by every vehicle during its simulation run, i.e., the sum of the individual values reported by each vehicle according to the formula stated earlier in Equation (Equation 1). The obtained results are shown in Figure 25 where at the two first vehicle densities, the total value increases from about 5 K to 18 K between 50 and 100 vehicle densities, respectively. The second half gives some remarkable increases in the obtained value, which starts from the last point, which is 18 K but jumps to about 45 K then jumps again to 70 K for the 150 and 200 vehicle densities, respectively. This indicates that beyond the density of 50 vehicles, the higher the density, the more road congestion events there will be, and hence the higher the road congestion value will be.

## 6. Discussion and Results Interpretation

To summarize the different obtained results and to give a general interpretation and discussion, we provide the following set of observations regarding the OVR scheme: the achieved location privacy level, QoS, congestion reporting and other remarks that make it easier to understand the strong and weak points of the OVR scheme.

With regard to the comparison between the five location privacy schemes, we summarized the resulting outcomes in both the Manhattan and highway scenarios in two tables: Table 5 and Table 6, respectively, and the higher the value, the better the scheme performances are. In each table, we selected the (1) privacy and (2) QoS metric classes, and for each class, we chose the atomic metrics mentioned previously in the performance evaluation section. They are presented in the same order as in the evaluation metrics subsection of the performance evaluation section, where $P_{x}$ is for the privacy-related metrics and $Q_{x}$ for the QoS metrics.

For more precise scores, we tended to assign weights to each one of the used atomic metrics, where a double weight was assigned to both the (1) normalized traceability metric because it is more significant and normalized, as stated previously and as pointed out in [41] and to the (2) total verified signatures since a single sent beacon triggers *k*-neighbor receptions, which is, as a result, a determining factor for QoS. Additionally, since the main focus of this research study is location privacy, we gave a double weight to the privacy metric class, and that gives a fair comparison between the five location privacy schemes. The last column in both tables indicates whether the scheme considers safety in its behavior, represented by a C-mark, or it does not, represented by an X-mark, respectively.

From the results, we can make a lot of observations and comments, but we will stick to the most important ones and narrow them down as follows:The location privacy evaluation results for both the Manhattan and highway scenarios show two major categories, $P_{1}$ and $P_{2}$ on the one hand, and $P_{3}$ and $P_{4}$ on the other. $P_{2}$ is considered meaningful and thus influences the overall scores, with SLOW being the best performing scheme in the Manhattan scenario, followed by OVR, and OVR being the best performing scheme in the highway scenario, followed by both CPN and RSP.for both of the Manhattan and highway scenarios in the QoS evaluation, CPN had a huge pseudonym consumption. This was due to the default *k* parameter used, which was 2, and hence $v_{i}$ triggers a pseudonym change once there are two neighboring vehicles. To mitigate this intense pseudonym consumption, two options are available: (1) using another privacy scheme or (2) changing the *k* parameter.CAPS and SLOW were not appropriate in the case of the highway scenario model, and it was due to the nature of these two schemes since no contexts were met by CAPS and no low speeds were conducted by SLOW and hence, no pseudonym changes were performed. Additionally, CAPS did consume fewer pseudonyms in the Manhattan scenario, followed by RSP, SLOW and OVR while in the highway scenario, RSP was the one with fewer pseudonym changes, followed by OVR.the scheme with less sent beacons in the Manhattan scenario was SLOW, followed by OVR, and in the highway scenario, it was OVR. The same interpretation applies for the less verified beacon signatures.for the last road congestion evaluation, despite OVR being a silent-period using scheme, it could overcome the resulting drawback which is the invisibility of silent vehicles existing on the different map roads. Using the density estimation mechanism and the report congestion protocol, OVR is able to estimate the existing density in congested areas and report the calculated values to the nearby static edge, which reduces the propagation delay and eliminates the need for the cloud server contact. This leads to a quasi-instant perception of road congestion by the existing vehicles.from an OVR-oriented perspective, the scheme clearly excels in the highway scenario traceability and normalized traceability in the privacy metric class and in the number of sent beacons and verified signatures in the QoS metric class.the privacy alone is not a determining factor for the privacy scheme choice; the safety factor is more important and this is the motivation behind the proposition of OVR in the first place. The safety feature was added in both of Table 5 and Table 6 and due to the fact that both of RSP and SLOW do not consider their silent mode entering moment, this will influence the comparison between the privacy schemes.the final thoughts of the different privacy and QoS evaluations in both scenarios were that OVR had outperformed the other schemes by taking the safety parameter into consideration, which is in fact much more important than both privacy and QoS.

## 7. Conclusions

The success of IoV deployment relies on many requirements, one of which is location privacy. In this study, we proposed OVR, a location privacy and road congestion scheme to deal with the privacy-safety trade-off resulting from the unstudied silent period using schemes. OVR exploits the three vehicle classes—normal, public transport, and role vehicles—to enhance their overall geo-location privacy level. The scheme covered the issue resulting from the “silent period” concept, namely the road congestion estimation, where it showed an estimation mechanism based on calculating the approximate number of vehicles between the delimiters. Furthermore, OVR proposed the exploitation of the mobile edge and static edge technologies to make fast road-congestion estimation and information spreading with low latencies for the real-time applications. Privacy and QoS metrics were used during the comparison of OVR versus the four privacy schemes, namely CAPS, CPN, RSP and SLOW where OVR showed better overall results for privacy and QoS, as drawn in two summarizing tables that take into account the results of the whole simulations and assign appropriate weights for each feature. The main contribution of OVR is the wise use of silent periods by exploiting the nature of vehicle types: (a) normal vehicles and (b) role and public transport vehicles to assert road safety, geo-location privacy and the road congestion estimation option.

Despite the beneficial use of OVR in terms of the privacy, QoS and safety, it may result in some limitations that should be addressed in the future. We briefly mention them as follows:the road congestion estimation is approximate but not absolute and the size of vehicles may differ in some scenarios.even in the privacy-seekers class, the option to give priority levels to some of the vehicles like VIPs and celebrities which need privacy the most to evade easy identification.the reliance on the nearby static edge servers brings forth the short transmission delays and dispenses with the need to contact the cloud server each time, yet the option for the edge servers may not be always available, and thus OVR can be used in situations of contacting the cloud server, which reduces the efficiency of the scheme.the situation of having vehicles that are not applying the OVR scheme may reduce the productivity of the scheme in terms of privacy, QoS and congestion reporting, and it can happen in two scenarios: (a) the start of IoVs integration and (b) the scenario of having different privacy schemes that are applied by the existing vehicles.

These later limitations can be covered in future works by giving enhancements and propositions to cope with the aforementioned gaps resulting from OVR.

## Figures and Tables

**Figure 1 sensors-23-00531-f001:**
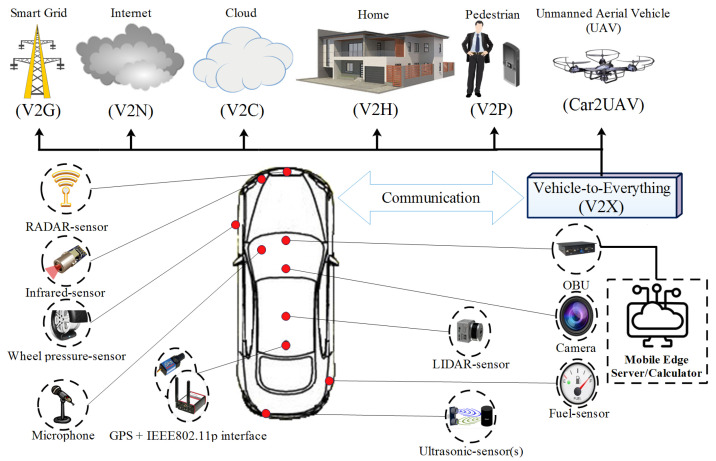
A conceptual illustration for the exploitation of embedded sensors in smart cars for environmental perception, the use of V2X technology for various communication types, and the deployment of OBU as a mobile edge server for IoV.

**Figure 2 sensors-23-00531-f002:**
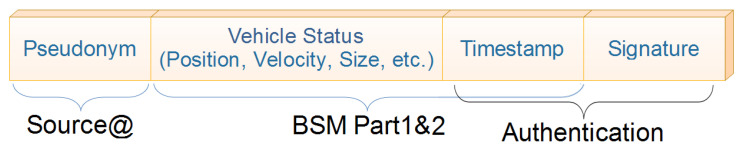
The basic safety message fields and format.

**Figure 3 sensors-23-00531-f003:**
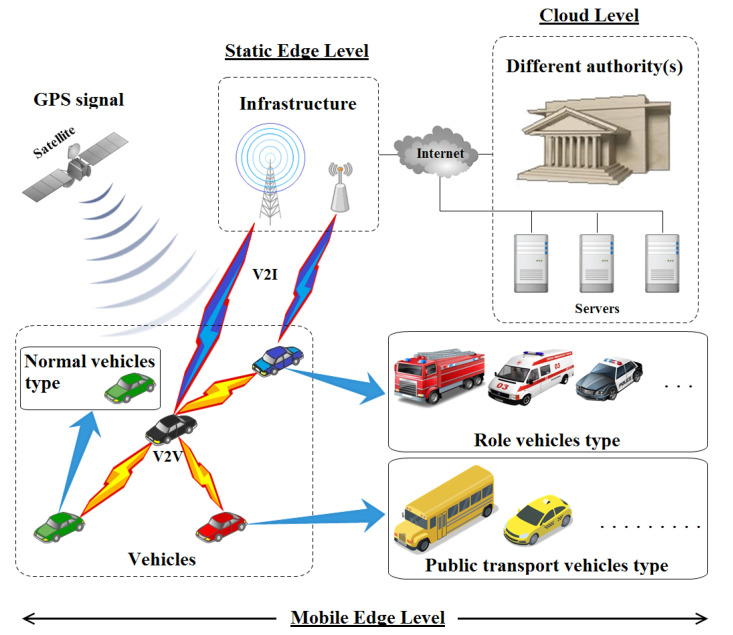
The supposed network model.

**Figure 4 sensors-23-00531-f004:**
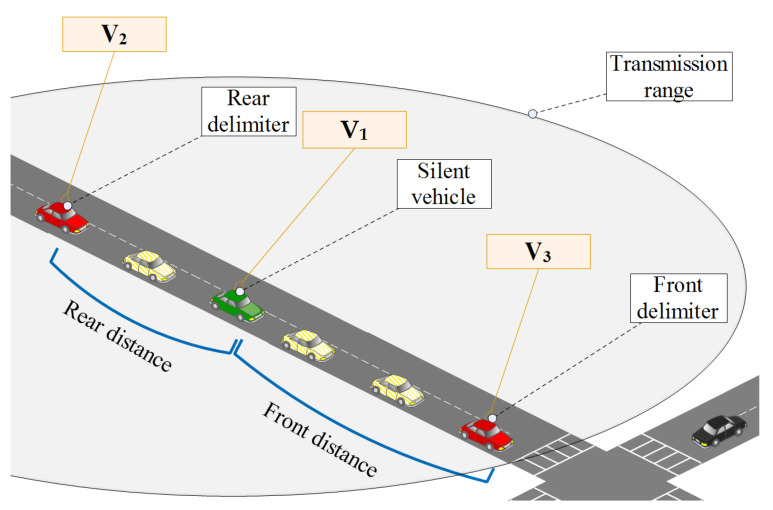
Proposed safety mechanism through front and rear delimiters.

**Figure 5 sensors-23-00531-f005:**
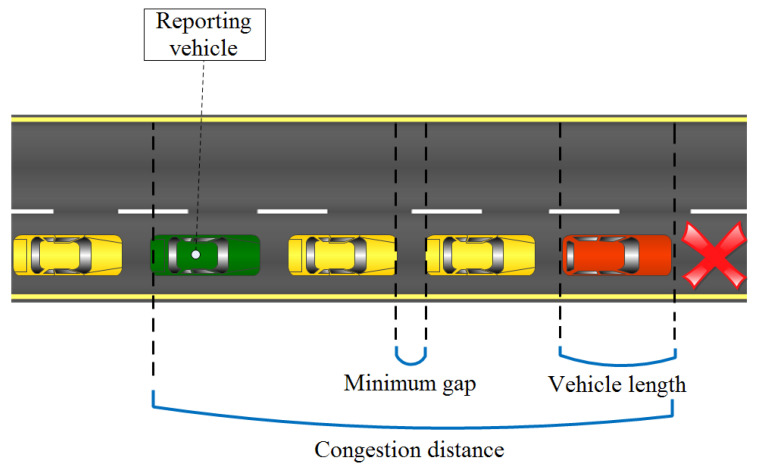
Density estimation mechanism.

**Figure 6 sensors-23-00531-f006:**
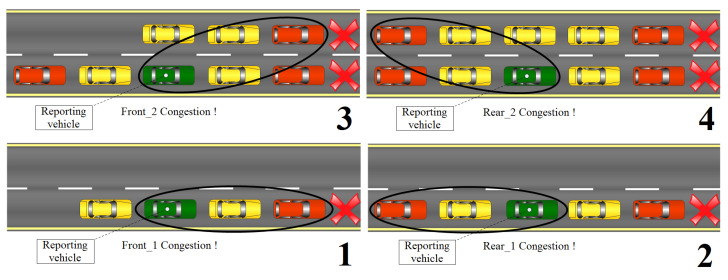
Road congestion reporting scenarios.

**Figure 7 sensors-23-00531-f007:**
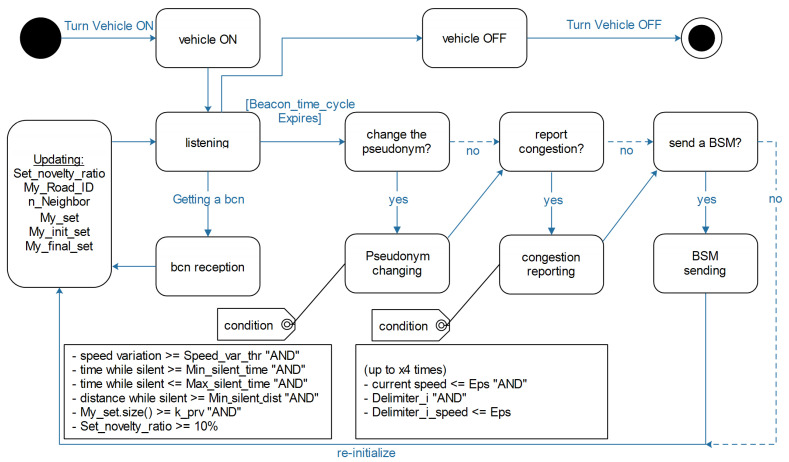
A summarizing state diagram of OVR.

**Figure 8 sensors-23-00531-f008:**
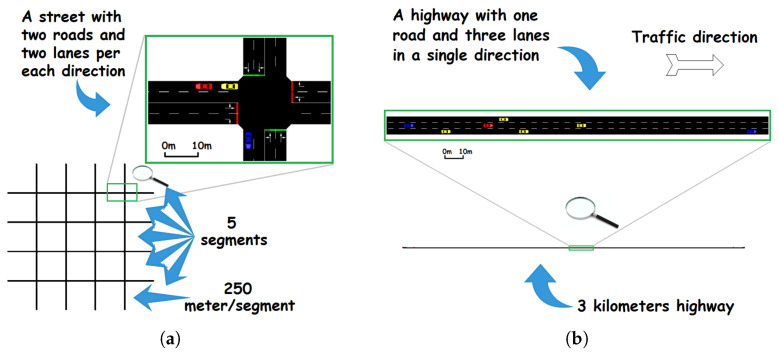
Simulation scenarios illustration. (**a**) The Manhattan scenario illustration. (**b**) The highway scenario illustration.

**Figure 9 sensors-23-00531-f009:**
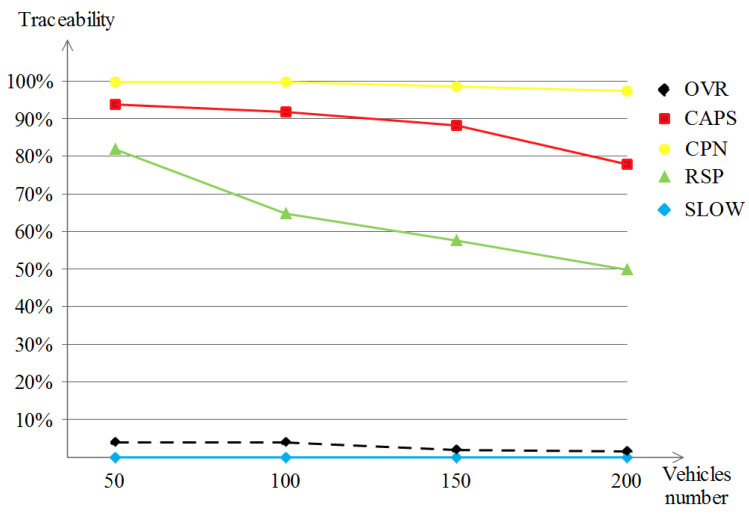
Traceability of privacy schemes in the Manhattan scenario.

**Figure 10 sensors-23-00531-f010:**
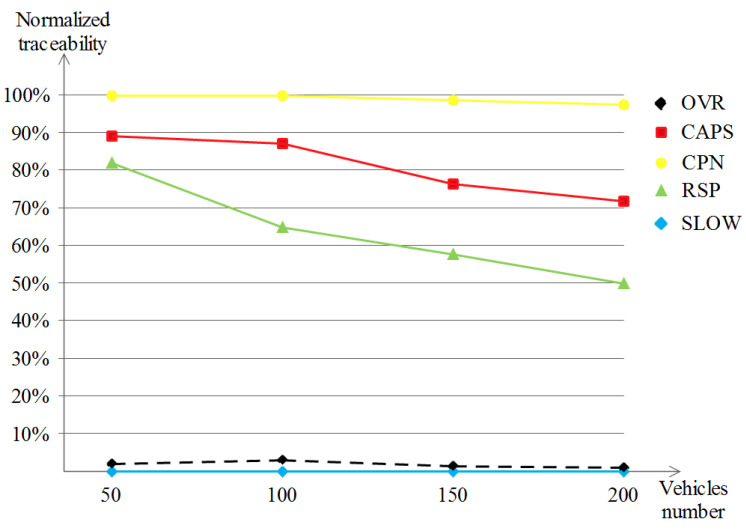
Normalized-traceability of privacy schemes in the Manhattan scenario.

**Figure 11 sensors-23-00531-f011:**
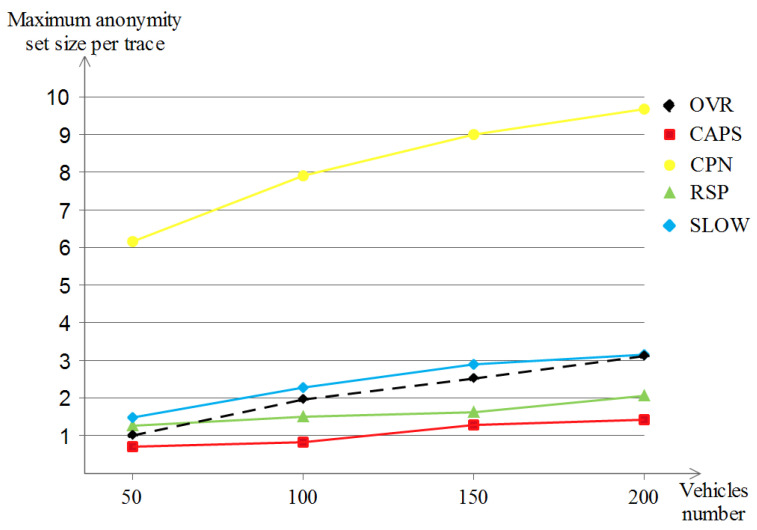
Maximum anonymity set size per trace of privacy schemes in the Manhattan scenario.

**Figure 12 sensors-23-00531-f012:**
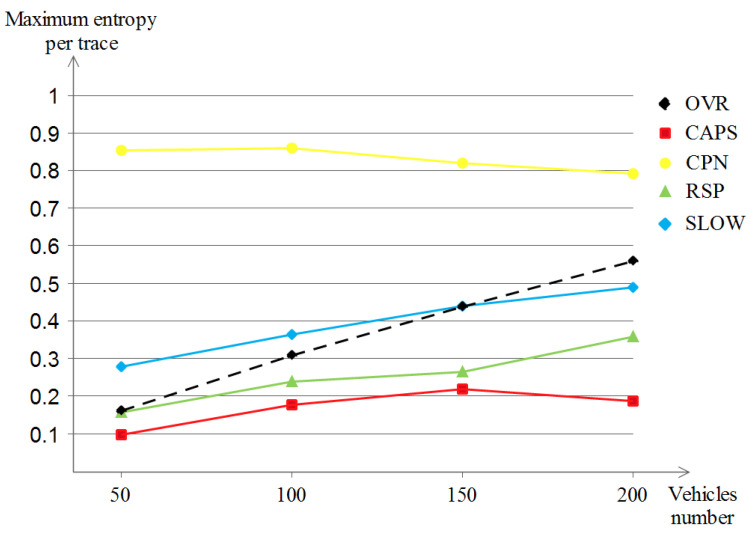
Maximum entropy per trace of privacy schemes in the Manhattan scenario.

**Figure 13 sensors-23-00531-f013:**
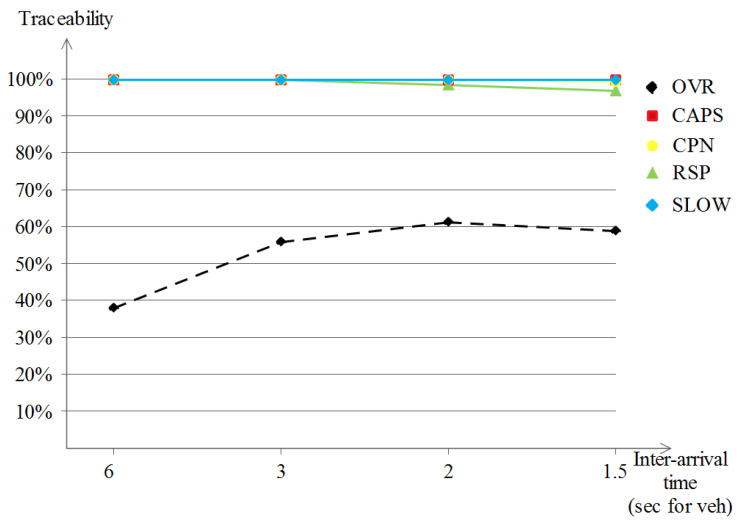
Traceability of privacy schemes in the highway scenario.

**Figure 14 sensors-23-00531-f014:**
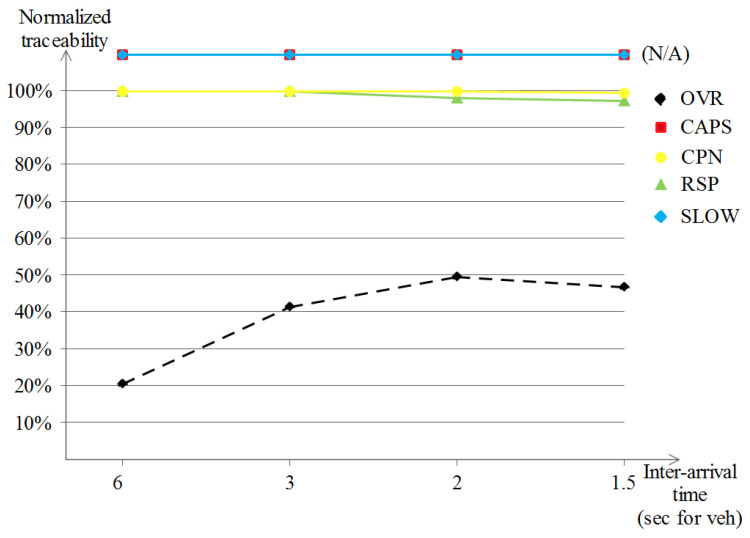
Normalized-traceability of privacy schemes in the highway scenario.

**Figure 15 sensors-23-00531-f015:**
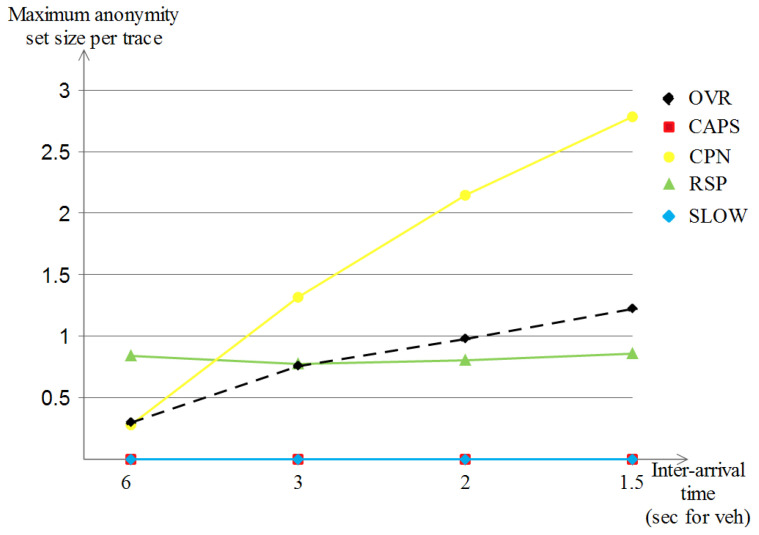
Maximum anonymity set size per trace of privacy schemes in the highway scenario.

**Figure 16 sensors-23-00531-f016:**
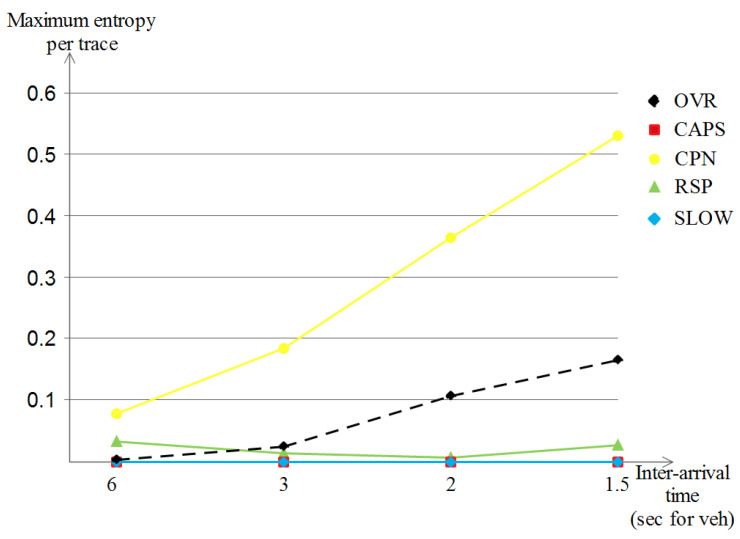
Maximum entropy per trace of privacy schemes in the highway scenario.

**Figure 17 sensors-23-00531-f017:**
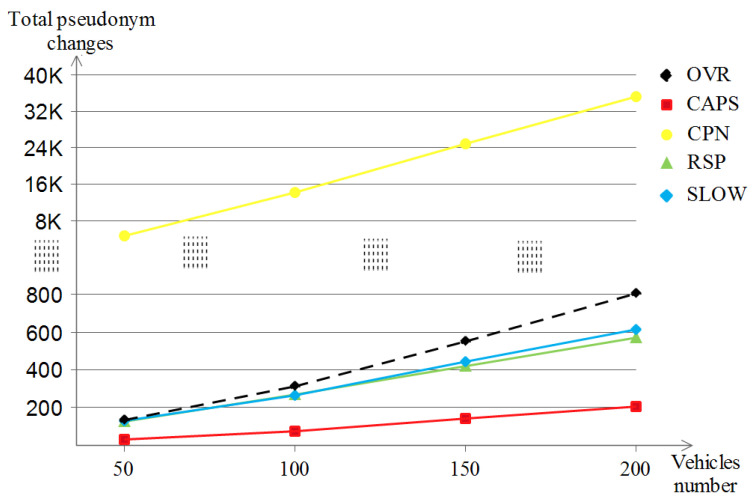
Pseudonyms consumption of privacy schemes in the Manhattan scenario.

**Figure 18 sensors-23-00531-f018:**
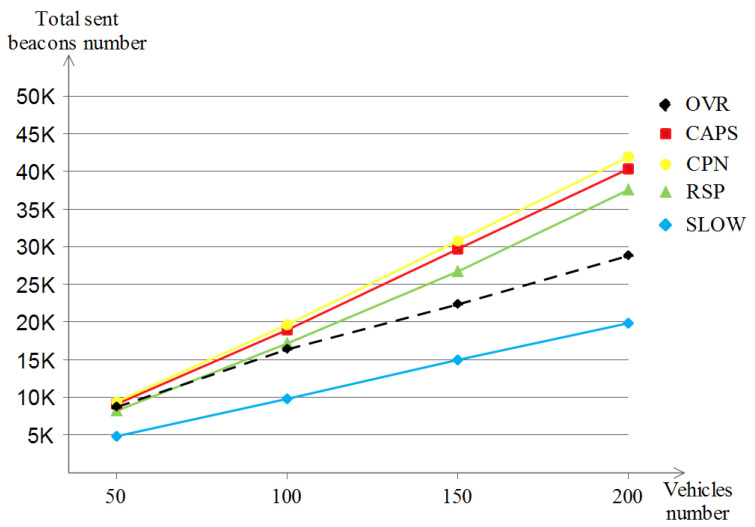
Sent beacons number of privacy schemes in the Manhattan scenario.

**Figure 19 sensors-23-00531-f019:**
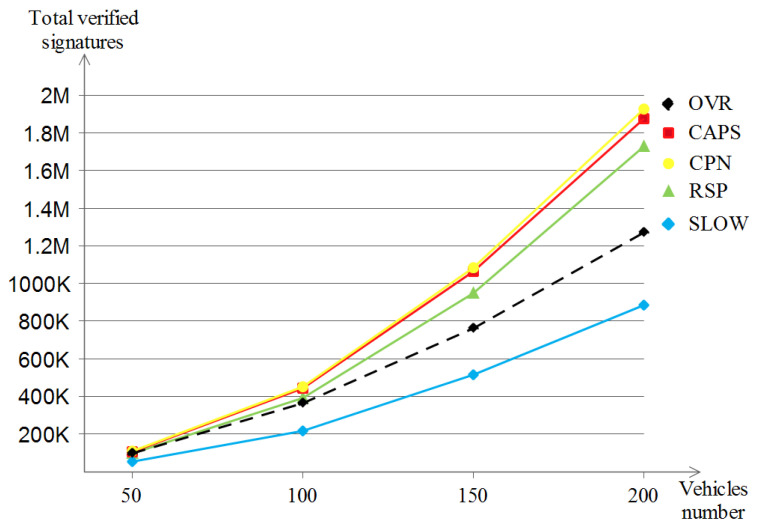
Verified signatures number of privacy schemes in the Manhattan scenario.

**Figure 20 sensors-23-00531-f020:**
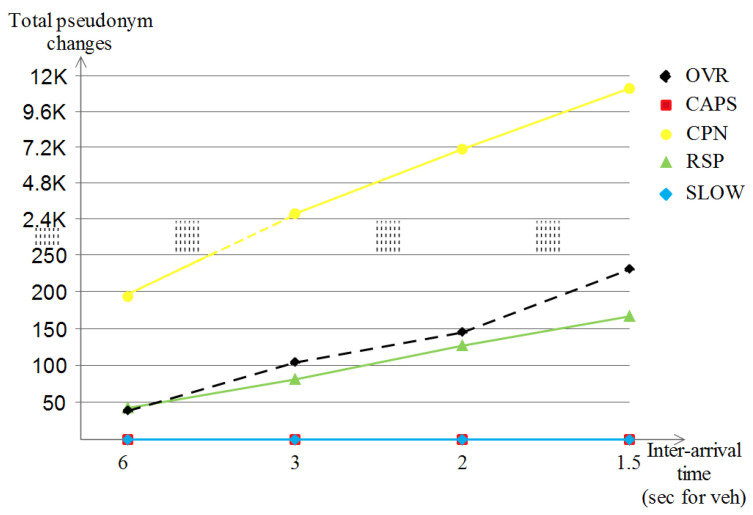
Pseudonyms consumption of privacy schemes in the highway scenario.

**Figure 21 sensors-23-00531-f021:**
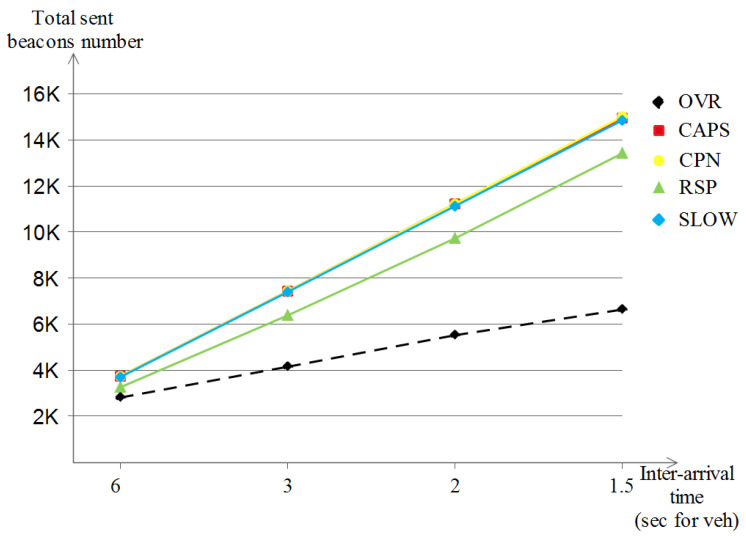
Sent beacons number of privacy schemes in the highway scenario.

**Figure 22 sensors-23-00531-f022:**
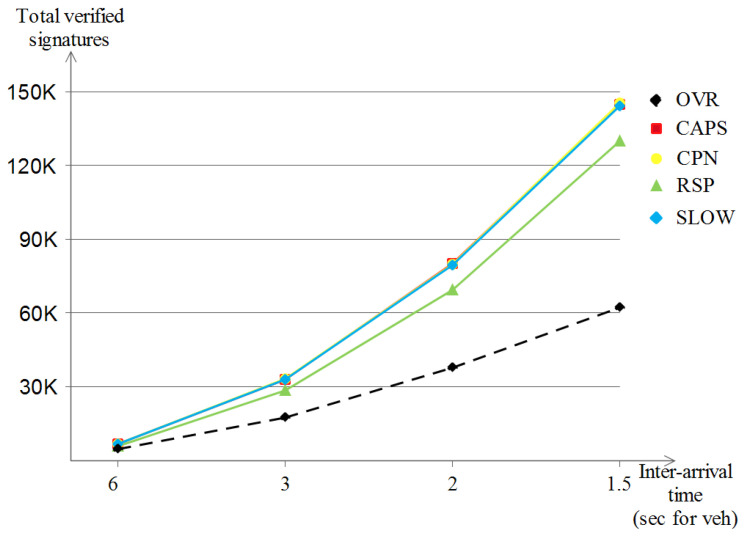
Verified signatures number of privacy schemes in the highway scenario.

**Figure 23 sensors-23-00531-f023:**
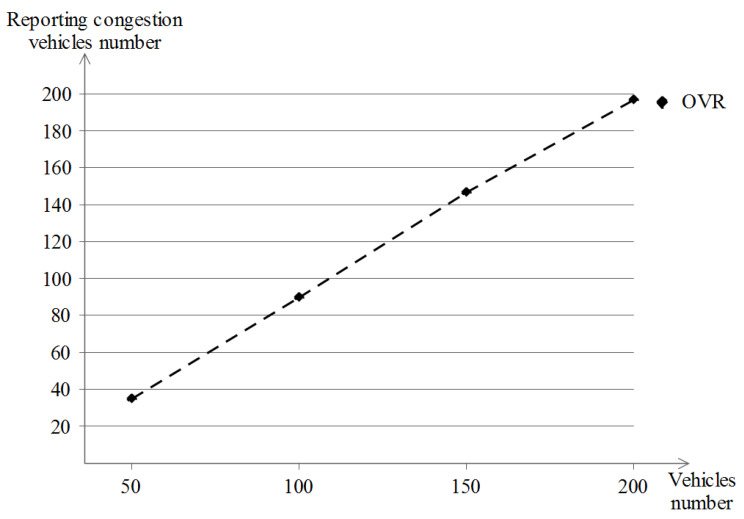
Number of vehicles that reported road congestion at least once in the Manhattan scenario.

**Figure 24 sensors-23-00531-f024:**
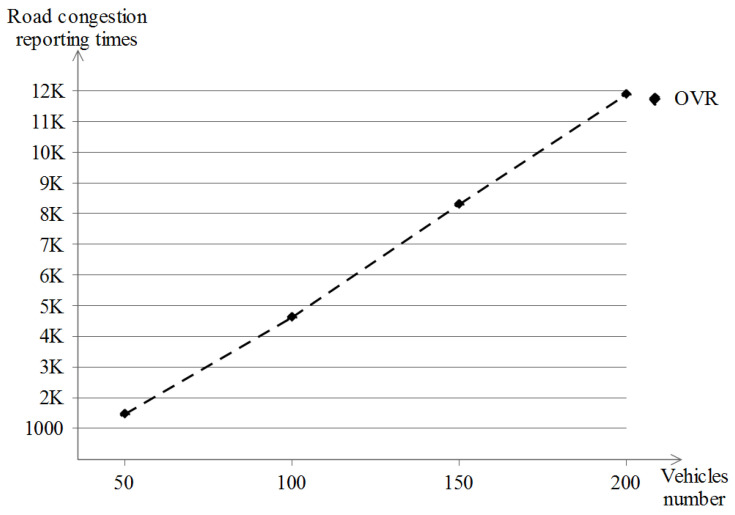
Number of times a road congestion was reported in the Manhattan scenario.

**Figure 25 sensors-23-00531-f025:**
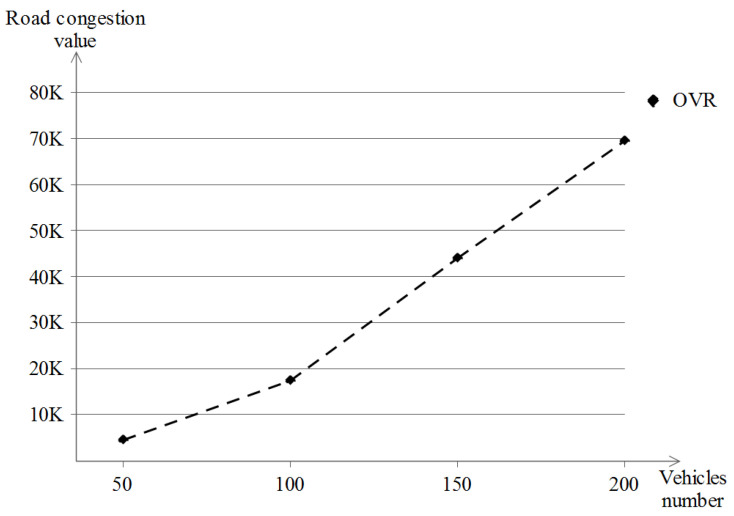
Value of the total road congestion in the Manhattan scenario.

**Table 1 sensors-23-00531-t001:** The used terminologies in OVR for the double variables.

Name	Unite	Role
Delimiter_radius	m	a fictive range that considers only vehicles v_j that are close with 200 m or below and at the same time are on the same road.
Prv_Neighbor_radius	m	a fictive range that adds vehicles v_j to a set of privacy vehicles myset (80 m).
Set_novelty_ratio	%	a percentage value between 0 and 100 that describes the ratio of novelty between the initial vehicles set and the final vehicles set.
Front_1_dist	m	the distance between v_i and the first/primary front delimiter.
Front_2_dist	m	the distance between v_i and the second/secondary front delimiter.
Front_1_speed	m/s	the speed of the primary front delimiter (in meter per second).
Front_2_speed	m/s	the speed of the secondary front delimiter (in meter per second).
Rear_1_dist	m	the distance between v_i and the first/primary rear delimiter.
Rear_2_dist	m	the distance between v_i and the second/secondary rear delimiter.
Rear_1_speed	m/s	the speed of the primary rear delimiter.
Rear_2_speed	m/s	the speed of the secondary rear delimiter.
Tempo_dist	m	the distance between v_i and v_j that is obtained from the received beacon message bcn.
Eps	m/s	a small value in which when driving below it, the congestion estimation starts working.
Min_speed	m/s	a variable to keep track of the minimum speed of v_i while it is in the privacy mode.
Max_speed	m/s	a variable to keep track of the maximum speed of v_i while it is in the privacy mode.
Speed_var_thr	m/s	a threshold for the speed variation in which the difference between Min_speed and Max_speed has to be greater.

**Table 2 sensors-23-00531-t002:** The used terminologies in OVR for the integer variables.

Name	Unite	Role
k_sft	number	the threshold number of vehicles considered to decide whether to enter silence or not (for safety).
k_prv	number	the threshold number of vehicles considered to decide whether to change the pseudonym or not (for privacy).
Car_type	number	the type of v_i (normal, role, public transport vehicles).
Front_1_id	number	the ID number of the primary front delimiter of v_i.
Front_2_id	number	the ID number of the secondary front delimiter of v_i.
Rear_1_id	number	the ID number of the primary rear delimiter of v_i.
Rear_2_id	number	the ID number of the secondary rear delimiter of v_i.
n_Neighbor	number	the number of neighbor vehicles from which v_i had received a beacon message before it sends its own.
Min_silent_time	second	the minimum period of time before the pseudonym change may take place.
Max_silent_time	second	the maximum period of time before the pseudonym change must take place.
Min_silent_dist	meter	the minimum distance before the pseudonym change may take place.
Max_silent_dist	meter	the maximum distance before the pseudonym change must take place.
Beacon_time_cycle	second	the consumed period of time before sending the next beacon.

**Table 3 sensors-23-00531-t003:** The used terminologies in OVR for the boolean variables.

Name	Unite	Role
Front_1	(available/not)	whether v_i has a primary front delimiter or not.
Front_2	(available/not)	whether v_i has a secondary front delimiter or not.
Rear_1	(available/not)	whether v_i has a primary rear delimiter or not.
Rear_2	(available/not)	whether v_i has a secondary rear delimiter or not.

**Table 4 sensors-23-00531-t004:** Simulation parameters and values.

	**Parameters**	**Values**
Mobility	Vehicles Insertion	Manhattan(M) = 50, 100, 150, 200 (vehs) Highway(HW) = 6, 3, 2, 1.5 (s/veh) Overseers percentage = 20% in M, 10% in HW
	Insertion method	Non-instant (consecutive insertions)
	Mobility Model	RandomTrips (SUMO model)
Environment	Used Map (first)	Manhattan grid model 4 roads, 250 (m) per segment
	Used Map (second)	Highway model 3 (km)
	Simulation Time	300 (s) for Manhattan and 60 (s) for highway
Evaluation	Privacy metrics	Traceability Normalized Traceability Maximum Anonymity Set Size per Trace Maximum Entropy per Trace
	QoS metrics	Total pseudonym changes Total sent beacons Total verified signatures
	Congestion metric	Road congestion value (for Manhattan)
Strategy	CAPS	Minimum Pseudonym lifetime = 60 (s) Maximum Pseudonym lifetime = 180 (s) Neighborhood Threshold = 50 (m) Silence period= from 3 up to 13 (s)
	CPN	Neighbors radius = 100 (m) Neighbors threshold = 2
	RSP	Pseudonym duration = 60 (s) Silence period= from 3 up to 11 (s)
	SLOW	Speed threshold = 8 (m/s) Silence threshold = 5 (s)
	OVR	Delimiter_radius = 200 (m) Prv_Neighbor_radius = 80 (m) Set_novelty_ratio = 10 (%) k_sft = 2 (m/s) k_prv = 2 (m/s) Silence period = from 7 up to 30 (s) Silence distance = 30 (m) and above Eps = 0.84 (m/s) Speed_var_thr = 0.84 (m/s)

**Table 5 sensors-23-00531-t005:** Schemes comparison in the Manhattan scenario.

	Privacy					QoS						
	$P_{1}$	$P_{2}$	$P_{3}$	$P_{4}$	$P_{t}$	$Q_{1}$	$Q_{2}$	$Q_{3}$	$Q_{t}$	Score	Rank	Safety
Weight	1	2	1	1	2	1	1	2	1			
CAPS	2	4	1	1	8	5	2	4	11	27	5	✓
CPN	1	2	5	5	13	1	1	2	4	30	4	✓
RSP	3	6	2	2	13	4	3	6	13	39	3	✗
SLOW	5	10	4	4	23	3	5	10	18	64	1	✗
OVR	4	8	3	3	18	2	4	8	14	50	2	✓

**Table 6 sensors-23-00531-t006:** Schemes comparison in the highway scenario.

	Privacy					QoS						
	$P_{1}$	$P_{2}$	$P_{3}$	$P_{4}$	$P_{t}$	$Q_{1}$	$Q_{2}$	$Q_{3}$	$Q_{t}$	Score	Rank	Safety
Weight	1	2	1	1	2	1	1	2	1			
CAPS	1	—	0	0	1	5	2	4	11	13	5	✓
CPN	1	2	4	4	11	1	1	2	4	26	3	✓
RSP	2	2	2	4	10	4	3	6	13	33	2	✗
SLOW	1	—	0	0	1	3	5	10	18	20	4	✗
OVR	3	3	3	6	15	2	4	8	14	44	1	✓

## Data Availability

The data presented in this study are available on request from the corresponding author.

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
