# Peer review of "A Safety-Aware Location Privacy-Preserving IoV Scheme with Road Congestion-Estimation in Mobile Edge Computing"

_sensors, 2023, doi:10.3390/s23010531_

Round 1
Reviewer 1 Report
Internet of Vehicles (IoV) paradigm is a promising paradigm; however, its deployment is still challenging due to many security and privacy issues like location tracking using overheard safety messages. This paper proposes a novel location privacy scheme that uses the silent period feature by letting the role vehicles ensure safety and allowing other vehicles to enter into silence mode, thus enhancing their location privacy.
The manuscript is generally well-written. On the other hand, the paper should be revised by considering the following issues:
MAJOR ISSUES
+Introduction section should be improved.
+The main contributions of the paper should be clearly given as a separate subsection in the introduction section.
+The organization of the paper should be clearly given as a separate subsection in the introduction section.
+ Many references in this paper are mostly recent publications (within the last 5 years) and relevant. On the other hand, it also includes outdated references (published more than one decade). The bibliography should be improved by adding most recent references.
+The figures/schemes are generally clear. They show the data properly. It is not difficult to interpret and understand them. On the other hand, Figure 1 should be explained better by adding more information to its caption.
+In Section 5, the titles of subsubsubsections should be given properly instead of 1\, 2\, 3\. (For example, "1\The number of consumed pseudonyms:" should be given more properly like "5.4.1.1.The number of consumed pseudonyms:")
+Figures should be drawn by using professional tools.
+ Section Numerical Results should be definitely improved. Figures should be clearly explained, especially giving more numerical details in the text/main body of the paper.
+ The conclusion should be improved by giving the key results and main contributions more clearly.
+ Future work part should be given in the conclusion section instead of discussion section.
MINOR ISSUES
+The grammatical errors and typos should be fixed.
+Size of Figure 1, 2, 3, 4, 5, and 6 should be increased.
+The authors should not leave much gap like the ones in page 5, 12, 14, 15, 17, 20, 24, 27, 29, 31.
+The references in the bibliography should be given in the same style. The following link should be checked: https://www.mdpi.com/authors/references
Author Response
Comment 1.0: Internet of Vehicles (IoV) paradigm is a promising paradigm; however, its deployment is still challenging due to many security and privacy issues like location tracking using overheard safety messages. This paper proposes a novel location privacy scheme that uses the silent period feature by letting the role vehicles ensure safety and allowing other vehicles to enter into silence mode, thus enhancing their location privacy. The manuscript is generally well-written. On the other hand, the paper should be revised by considering the following issues: (MAJOR ISSUES from 1.1 to 1.10, MINOR ISSUES from 1.11 to 1.14).
Comment 1.1: Introduction section should be improved.
Response 1.1: Done. Please see the updated introduction section.
Comment 1.2: The main contributions of the paper should be clearly given as a separate subsection in the introduction section.
Response 1.2: Done. We put the main contributions of the paper in a separate subsection and made some adjustments to that part.
Comment 1.3: The organization of the paper should be clearly given as a separate subsection in the introduction section.
Response 1.3: Done. Please see the updated introduction section.
Comment 1.4: Many references in this paper are mostly recent publications (within the last 5 years) and relevant. On the other hand, it also includes outdated references (published more than one decade). The bibliography should be improved by adding most recent references.
Response 1.4: Done. We reinforced the bibliography of the manuscript with more recent publications (see the references below).
- Al-Shareeda, M.A.; Manickam, S.; Mohammed, B.A.; Al-Mekhlafi, Z.G.; Qtaish, A.; Alzahrani, A.J.; Alshammari, G.; Sallam, A.A.; Almekhlafi, K. Cm-cppa: Chaotic map-based conditional privacy-preserving authentication scheme in 5g-enabled vehicular networks. Sensors 2022, 22, 5026.
- Xu, C.; Ding, Y.; Chen, C.; Ding, Y.; Zhou, W.; Wen, S. Personalized location privacy protection for location-based services in vehicular networks. IEEE Transactions on Intelligent Transportation Systems 2022.
- Elahi, M.M.; Rahman, M.M.; Islam, M.M. An efficient authentication scheme for secured service provisioning in edge-enabled vehicular cloud networks towards sustainable smart cities. Sustainable Cities and Society 2022, 76, 103384.
Comment 1.5: The figures/schemes are generally clear. They show the data properly. It is not difficult to interpret and understand them. On the other hand, Figure 1 should be explained better by adding more information to its caption.
Response 1.5: Done. We made the caption be more descriptive by changing it into “A conceptual illustration for the exploitation of embedded sensors of smart cars for environmental perception, the use of V2X technology for various communication types and the deployment of OBU as a mobile edge server for IoV”.
Comment 1.6: In Section 5, the titles of subsubsubsections should be given properly instead of 1\, 2\, 3\. (For example, "1\The number of consumed pseudonyms:" should be given more properly like "5.4.1.1.The number of consumed pseudonyms:").
Response 1.6: Done. The format of such titles was corrected by adding an automatic enumeration by increasing the latex counter depth.
Comment 1.7: Figures should be drawn by using professional tools.
Response 1.7: We apology for the quality of figures and graphs. The tool that we used was EdrawMax with few customized items. We adjusted the size of some figures with the aim of having better graphical figures.
Comment 1.8: Section Numerical Results should be definitely improved. Figures should be clearly explained, especially giving more numerical details in the text/main body of the paper.
Response 1.8: Done. Numerical details were given to have better results interpretations. See the updated results section.
Comment 1.9: The conclusion should be improved by giving the key results and main contributions more clearly.
Response 1.9: Done. We extended the conclusion section to include the research characteristics and main contributions; find them in the updated conclusion section please.
Comment 1.10: Future work part should be given in the conclusion section instead of discussion section.
Response 1.10: Done. We moved the future work part into the appropriate location which is the conclusion section.
Comment 1.11: The grammatical errors and typos should be fixed.
Response 1.11: Done. We reviewed the article aiming at correcting the grammatical errors and typos. See the updated version of the manuscript. The corrections were written in blue color.
Comment 1.12: Size of Figure 1, 2, 3, 4, 5, and 6 should be increased.
Response 1.12: Done. The size of these figures was increased accordingly. Additionally, Figure 4 was removed since it does not give many important additions and Figure 5 was slightly modified to only show the important entities.
Comment 1.13: The authors should not leave much gap like the ones in page 5, 12, 14, 15, 17, 20, 24, 27, 29, 31.
Response 1.13: Done. The latex auto-generated gaps were fixed in the revised manuscript version. Also, since the manuscript is not in its final shape, the actual float-environment of figures may leave some gaps. A fix would be to adjust the float-environment parameters to better position the figures in question.
Comment 1.14: The references in the bibliography should be given in the same style. The following link should be checked: https://www.mdpi.com/authors/references.
Response 1.14: Some references are new and hence; some details are missing like the volume and number. A revision for the whole references was made to fix as many references as possible.
Reviewer 2 Report
There are many problems in this Article, which can be summarized as follows:
1. In the Introduction, the contribution of the article is not clear, the premise of the reader's recognition of the author's work is the need to have experimental proof, rather than simple theoretical analysis.
2. The author's creative research is based on a large number of related work. What are the author's outstanding contributions compared with these work?
3. The author uses a social feature to improve the overall level of location-privacy. However, does the author consider that this information is not only used to locate the vehicle location, but also needs to be used for vehicle unique identification and vehicle communication. Without unique identification, how can the communication between the V2V, V2P, V2N, etc.?
4. The organization of the Related Work is disorganized. What does the author mean? How do these jobs relate to yours?
5. On page 7, the author mentions that "role vehicles may be utilized for boosting-up the normal vehicles' privacy", "such public transport vehicles may also be referred with the name of "safety-providers"". Is the scheme proposed by the author reasonable? Is it easier for attackers to obtain location privacy information if they disguise themselves as role vehicles or public transport vehicles?
6. Figure 4 has little practical meaning and takes up a lot of space.
7. Figure 5 has a similar problem as Figure 4.
8. The author's experiments are rich in content, but the organization and logic are chaotic, and the readers do not know what experimental results and arguments the author wants to express.
Author Response
Comment 2.0: There are many problems in this Article, which can be summarized as follows:
Comment 2.1: In the Introduction, the contribution of the article is not clear, the premise of the reader's recognition of the author's work is the need to have experimental proof, rather than simple theoretical analysis.
Response 2.1. We reformulated and segmented the introduction for a better readability. Please, see the updated introduction part. The experimental proof is indeed missing since we had chosen to validate our work by simulation means and some theoretic analysis. The reason behind the choice of the validation method is the lack of available experimental resources, yet we made sure to choose credible simulators, i.e., SUMO, Veins and PREXT.
Comment 2.2. The author's creative research is based on a large number of related work. What are the author's outstanding contributions compared with these work?
Response 2.2. We had modified the manuscript to make the outstanding contributions be more comprehensive. The “motivation” part of the introduction is dedicated to better describe what is missing in the related work.
Comment 2.3. The author uses a social feature to improve the overall level of location-privacy. However, does the author consider that this information is not only used to locate the vehicle location, but also needs to be used for vehicle unique identification and vehicle communication. Without unique identification, how can the communication between the V2V, V2P, V2N, etc.?
Response 2.3. We totally agree with the reviewer. The thing is, in this research, we based on communications that are based on BSM messages which are broadcasted to the next hope and thus, this kind of applications can have dedicated identifier / pseudo-identifier. It was already discussed in earlier works in which researchers had mentioned the necessity to change the identifiers of the whole communications stack (e.g., mac address, etc.). Thus, it is a trade-off between privacy and QoS and communications smoothness as mentioned in the related work section in the work of Schoch et al. (2006). Thanks for this precise observation, we had emphasized this information in our revised manuscript version: “Changing one communication identifier is generally not enough because there are other identifiers in the same communication stack that are not changed like mac address and hence, such identifiers also have to be changed but this change has to be conducted carefully in order not to affect the other communications like V2P, V2C, etc.”
Comment 2.4. The organization of the Related Work is disorganized. What does the author mean? How do these jobs relate to yours?
Response 2.4. The related work section does indeed contain a bunch of what looks like random related work. The elaborated related work section has three main aims: (1) introduces the concepts used to achieve geo-location privacy like the change of pseudonyms and the use of silent periods, (2) gives more details about the compared schemes and how do they operate and (3) pin some other geo-location privacy schemes so that the reader will be aware of.
Comment 2.5. On page 7, the author mentions that "role vehicles may be utilized for boosting-up the normal vehicles' privacy", "such public transport vehicles may also be referred with the name of "safety-providers"". Is the scheme proposed by the author reasonable? Is it easier for attackers to obtain location privacy information if they disguise themselves as role vehicles or public transport vehicles?
Response 2.5. If the attackers disguise themselves as role vehicles or public transport vehicles, they would definitely affect the location privacy if they decide to enter the silent period mode; forcing normal vehicles to achieve road safety by quitting the silence mode and going into the safety-achiever mode. However, mere eavesdropping or data collection by disguised attackers has no direct impact on privacy information because BSMs are already sent in clear with light authentication and signature mechanisms.
Comment 2.6. Figure 4 has little practical meaning and takes up a lot of space.
Response 2.6. We agree. Figure 4 was removed since the given information does not require a whole figure for that.
Comment 2.7. Figure 5 has a similar problem as Figure 4.
Response 2.7. Done. Now, only the important entities in addition to the concepts used in OVR are kept. Few information is meant by figure 5 like the front and rear distances’ exact beginning and end point, that is used in the protocols of OVR.
Comment 2.8. The author's experiments are rich in content, but the organization and logic are chaotic, and the readers do not know what experimental results and arguments the author wants to express.
Response 2.8. We updated the manuscript and emphasized the motivation and the contributions that this research had provided. In addition, some parts were restructured for better readability and comprehension.
Round 2
Reviewer 1 Report
The authors addressed my comments sufficiently so the paper is acceptable in its current form. Minor spell check is suggested.
Reviewer 2 Report
It has relevant reference value in the field of IoV.